# DPO Unchained: Your Training Algorithm is Secretly Disentangled in Human Choice Theory (and its Loss' Convexity is Dispensable)

Wenxuan Zhou [1]  Shujian Zhang [1]  Brice Magdalou [2]  John Lambert [1]  Ehsan Amid [3]  Richard Nock [4]
Andrew Hard [1]

## Abstract

Normative theories allow one to elicit key parts of a ML algorithm from first principles, which is crucial at a time of championed scrutiny for ML work. Direct Preference Optimization (DPO) cleverly bypasses reward modeling by making an explicit link with a specific normative model of human choice. Our paper elevates this connection to the full generality of DPO's normative framework. Getting there requires reworking social choice theory's textbook path for a better RLHF/ML fit. It elevates the connection to a remarkably broad viewpoint on preference optimization, considering the current panorama of DPO follow-ups. It also unveils unexpected riches for ML, chief among which the support for *non-convex* losses, the fact that *any* compliant ML analytical choice can be embedded with *any* human choice model, and a normative framework's umbrella wide enough to safeguard DPO's *extensions* (margins, length correction, ...). A *toy* experiment "far away" from the DPO crowd is given.

## 1. Introduction

At a time when machine learning (ML) faces increasing scrutiny on its technical contributions (Schaeffer et al., 2025), normative theories constitute a solid bedrock of ML. For example, they allow one to elicit the starting point of any ML algorithm, the objective function to be optimized, from first principles. Bregman divergences (Reid & Williamson, 2010; Williamson et al., 2016), $f$-divergences (Csiszár, 1991), optimal transport distances (Peyré & Cuturi, 2019; Villani, 2009) all follow from first principles built on

two pillars: analytical (mathematical or physics grounding) and rational (modeling human behavior). ML is a poster child of a related phenomenon whereby elegant and very successful breakthroughs exploit a *minuscule* fraction of the normative freedom while (i) colossal pressure is put for their fast generalization, (ii) challenges mount on their normative bedrock, yet (iii) the full extent of the normative bedrock and the ML freedom it authorizes are completely unknown!

Case in point: Direct Preference Optimization. DPO introduced a trick to bypass reward modelling in Reinforcement Learning from Human Feedback (RLHF), which trains reward function $r_\omega$ then policy $\pi_\theta$ with parameters $\theta$. RLHF makes use of Bradley-Terry-Luce's (BTL) model (Bradley & Terry, 1952) in its loss formulation, but its core normative safeguard to learn the reward is Savage's properness theory (Savage, 1971). DPO bypasses learning the reward function parameters ($\omega$) by introducing an explicit direct link $r_\theta = f(\pi_\theta)$ using the BTL model: in doing so, it introduces a normative safeguard from human choice theory.

Savage's framework is as general as BTL is specific since BTL *only* supports DPO's specific choice for the link $r_\theta = f(\pi_\theta)$. DPO's popularity soared as fast as it became clear that its BTL component was increasingly playing the role a straitjacket for algorithm's design, some opting to remain very closely to BTL (Chen et al., 2025b; Zhang et al., 2025), others explicitly abandoning any human choice affiliation (Khorasani et al., 2025). It is hard to exaggerate the risk of jeopardizing the human choice component in the current context of applications of LLMs, even more in the broader ML space (Schaeffer et al., 2025). Yet, the knowledgeable eye inevitably remarks that a *second* normative component fully supports the rest of DPO's analytical choices, i.e. its starting point, the RLHF objective, and its final loss, the cross-entropy: Savage's properness. In the context of DPO and the challenges at stake developed above, that these two fundamental normative theories meet at such a successful approach **inevitably begs a two-part question**:

(on the normative part) *how can one conveniently generalize BTL in such a way that DPO's connection generalizes to the entirety of Savage's framework, and* (on the ML part) *what is the freedom this general framework brings to design*

(Listing order is random)  [1]Google DeepMind [2]CEE-M, Univ Montpellier, CNRS, INRAE, Institut Agro [3]Work done while at Google DeepMind [4]Google Research. Correspondence to: Richard Nock <richardnock@google.com>.

*Proceedings of the 43rd International Conference on Machine Learning*, Seoul, South Korea. PMLR 306, 2026. Copyright 2026 by the author(s).

*preference optimization algorithms "à-la-DPO" ?*

While the second part is obviously important for ML, the first part is also important because it explains the *operating frameworks* behind the scenes for preference optimization. It can also influence the design of human experiments.

**Our paper addresses and answers both parts** (and more). Specifically,

- *on the normative part*, we build upon the seminal framework of Doignon & Falmagne (1974), broadly generalizing BTL. However, a proper embedding in the ML/RLHF framework requires bypassing the use of its core axiom, solvability: we do it by introducing lotteries fully compatible with ML/RLHF's sampling, built on top of two seminal pillars of normative economics from Machina and Davidson/Marschak (Machina, 1985; Davidson & Marschak, 1958). The normative framework we end up with unveils a third "dimension" to RLHF's diptych (choice, preference) with the possibility to *abstain* from picking a choice. The *normative structure* of abstention is of independent interest;
- *on the ML part*, what constitutes the analytical part of the algorithm – two functions, one for the rewards, one for the loss, as in DPO – stems from three independent bricks, two proper losses and one function defining human choice;
- *on the freedom-of-design part*, the freedom that this general normative bedrock authorizes is nothing short of considerable, and quite unexpected in places:

  1. our main result shows that any valid choice of the two-part algorithms functions (rewards / loss) can fit with **any** of our human choice components. Hence, there is absolutely no need to tie / justify a preference optimization algorithm with a specific human choice model (which constrains analytical choices, as seen from the numerous follow-ups to DPO calling to BTL);
  2. a normative bedrock inevitably imposes regularity constraints on the analytical parts: the convexity of losses for real-valued prediction (supervised learning with e.g. the logistic loss) follows from Savage's properness. Our main result **alleviates** the need for convexity: preference optimization also operates with non-convex losses;
  3. successful extensions of DPO have been designed, for home advantages, margins and length normalization (Meng et al., 2024; Park et al., 2024; Li et al., 2025; Zhao et al., 2025). We show that our normative framework also **covers** and safeguards those, so ultimately it is not restricted to preference optimization "à-la-DPO";

We also propose some toy experiment on how our framework can be used to design preference optimization algorithms that radically depart from the state of the art, still pretty much in a small world around DPO if we consider the considerable map for preference optimization that we unveil. All proofs and additional details appear in an Appendix.

## 2. Basic definitions

We adopt many definitions from Rafailov et al. (2023). States / contexts / prompts are denoted by $x_{(.)} \in \mathcal{X}$, actions / alternatives / answers by $y_{(.)} \in \mathcal{Y}$. We denote for short $n \doteq |\mathcal{Y}|$ and $m \doteq |\mathcal{X}|$, $|.|$ being the size. We sometimes enumerate the elements of $\mathcal{Y}$ in $\mathbb{N}$ as $y_1, y_2, ..., y_n$. $\Delta_{\mathcal{Y}}$ being the set of distributions over $\mathcal{Y}$, a policy $\pi$ is an element of $\Delta_{\mathcal{Y}}^{\mathcal{X}}$, associating to each state a distribution over actions. There exists a reference (pretrained) policy, $\pi_{\text{ref}}$. We also abbreviate the probability simplex $\Delta_n$ for short.

### 2.1. Direct Preference Optimization (DPO)

Given a reward function $r : \mathcal{X} \times \mathcal{Y} \to \mathbb{R}$, Reinforcement Learning from Human Feedback (RLHF) then seeks a policy $\pi_\theta \in \Delta_{\mathcal{Y}}^{\mathcal{X}}$ maximizing the per-state objective ($\forall x \in \mathcal{X}$):

$$J_{\text{KL},r}(\pi_\theta(.|x))$$
$$= \mathbb{E}_{y \sim \pi_\theta(.|x)}[r(x,y)] - D_{\text{KL}}(\pi_\theta(.|x) \| \pi_{\text{ref}}(.|x)). (1)$$

To get to DPO, one first remarks that the rewards satisfy at the optimum of (1) $r_\theta(x,y) = \log \frac{\pi_\theta(y|x)}{\pi_{\text{ref}}(y|x)} + Z(x)$. Second, Bradley-Terry-Luce's model builds choice probabilities from rewards ($\sigma(z) \doteq 1/(1 + \exp(-z))$ is the sigmoid):

$$p_\theta(y \succ y'|x) \quad \doteq \quad \sigma(r_\theta(y,x) - r_\theta(y',x)). \quad (2)$$

Disregarding parameter $\theta$ and the underlying measure, $p(y \succ y'|x)$ is the probability that $y \in \mathcal{Y}$ be *chosen* over $y' \in \mathcal{Y}$ given $x \in \mathcal{X}$ (the fundamental notion of *preference* is defined later).

Third and last, there remains to include those probabilities in a loss for class-probability estimation which can be directly optimized from user's data. In the case of DPO, it is the log-loss $I(\pi_\theta) = \mathbb{E}_{(x,y,y') \sim \mathcal{D}} [- \log p_\theta(y \succ y'|x)]$. Importantly, *observation* $(x, y, y')$ means that $y \in \mathcal{Y}$ is *chosen* over $y' \in \mathcal{Y}$ given $x \in \mathcal{X}$. The final loss to be optimized only depends on policies and has a well-known shape:

$$I(p_\theta) = \mathbb{E}_{(x,y,y') \sim \mathcal{D}}\left[\psi\left(-\left(\log\frac{\pi_\theta(y|x)}{\pi_{\text{ref}}(y|x)} - \log\frac{\pi_\theta(y'|x)}{\pi_{\text{ref}}(y'|x)}\right)\right)\right], (3)$$

where $\psi(-z) \doteq \log(1 + \exp(-z))$ is the logistic loss.

### 2.2. Bregman divergences

It is remarkable (but, as we shall see, not unexpected) that all the blue parts above are tightly linked: the sigmoid is the inverse canonical link of the log-loss, the logistic loss is its dual convex surrogate (Nock & Nielsen, 2008; Reid & Williamson, 2010) and finally the KL divergence is the *Bregman divergence* of the log-loss. A Bregman divergence $D_{\phi,\boldsymbol{H}}(\boldsymbol{u}\|\boldsymbol{v}) \doteq \phi(\boldsymbol{u}) - \phi(\boldsymbol{v}) - (\boldsymbol{u} - \boldsymbol{v})^\top \boldsymbol{H}(\boldsymbol{v})$ is defined for convex $\phi$ and a selection of its subdifferential, $\boldsymbol{H} \in \partial\phi$. Bregman divergences have a second form, introduced in

information geometry by Amari & Nagaoka (2000) (see also Blondel et al. (2020)), whereby $D_\phi(\boldsymbol{u}\|\boldsymbol{v}^*) \doteq \phi(\boldsymbol{u}) + \phi^\star(\boldsymbol{v}^*) - \boldsymbol{u}^\top\boldsymbol{v}^*$, where $\varphi^\star(\boldsymbol{v}) \doteq \sup_{\boldsymbol{t}} \boldsymbol{t}^\top\boldsymbol{v} - \varphi(\boldsymbol{t})$ is the convex conjugate. This representation keeps implicit the dependence on the subgradient.

## 3. Bedrock

### 3.1. On loss functions

A loss function is a function $L(\boldsymbol{p},\boldsymbol{q}) : \Delta_n \times \Delta_n \to \mathbb{R}$.

**Definition 3.1.** Fix $\boldsymbol{p} \in \Delta_n$. Loss $L$ is $\boldsymbol{p}$-proper iff $L(\boldsymbol{p},\boldsymbol{q}) \geq L(\boldsymbol{p},\boldsymbol{p}), \forall \boldsymbol{q} \in \Delta_n$. It is proper iff it is $\boldsymbol{p}$-proper for any $\boldsymbol{p} \in \Delta_n$.

*Strict* properness holds if the inequality is an equality only for $\boldsymbol{q} = \boldsymbol{p}$. The log-loss of DPO is strictly proper. Properness matters because it encourages guessing the ground truth. Savage proposed to use an appealing form for $L$:

$$L(\boldsymbol{p},\boldsymbol{q}) \quad \doteq \quad \boldsymbol{p}^\top\boldsymbol{\ell}(\boldsymbol{q}) = \mathbb{E}_{i\sim\text{CAT}(\boldsymbol{p})}[\ell_i(\boldsymbol{q})], \qquad (4)$$

where "CAT" denotes a categorical distribution with parameter $\boldsymbol{p}$. Any proper loss is concave, but admits via convex duality an "equivalent" convex loss for real-valued prediction: DPO's logistic loss in (3) is the dual of its log-loss. Properness is intensional (Walder & Nock, 2020) and can be tricky to operate if (parts of) the loss itself need to be learned (Alfano et al., 2025; Gupta et al., 2025). Fortunately, there is a characterization of proper losses that makes handling a lot easier: all proper losses can be represented as Bregman divergences, and reciprocally every Bregman divergence "encodes" (the regret of) a proper loss (Nock & Nielsen, 2008; Williamson et al., 2016). RLHF's KL divergence in (1) is the Bregman representation of the log-loss.

### 3.2. On human choice and preferences

Savage's properness theory is very broad compared to Bradley-Terry-Luce's model (BTL) so we need to get "above" BTL to make a complete connection with the full extent of Savage. BTL is a stochastic (human) choice model that determines, when comparing two options $y_1, y_2 \in \mathcal{Y}$, the probability of $y_1$ being chosen over $y_2$, computed as the reward obtained from $y_1$, normalized by the sum of the two rewards. An important property is that, for a given prompt $x \in \mathcal{X}$, we have $p(y_1 \succ y_2|x) + p(y_2 \succ y_1|x) = 1$, so that abstention is not allowed. Nevertheless, because the set of options $\mathcal{Y}$ can constitute a highly heterogeneous choice space, certain pairs may simply prove to be incomparable to a human evaluator.

We introduce a generalization of BTL that addresses the problem of pairwise choice in a local manner, which we then extend globally much like a graph: it contains chains of completely comparable options, but also pairs of incompara-

ble ones (resulting in abstention). The appropriate axiomatic framework for deriving such a model is that proposed by Doignon & Falmagne (1974), or Krantz et al. (1989) (Section 17.2.3, Theorem 2). However, this approach relies on a specific axiom called "solvability" (Debreu, 1958), which imposes an uncountable set of actions and is therefore unfit for our purpose. To bypass this issue, we first assume that choice probabilities arise from an explicit maximization problem over "lotteries", as proposed by Machina (1985). An example of lottery we use is $(y_1 y_2)_\alpha$ ($\alpha \in [0, 1]$), where one of the two options is revealed to the human evaluator through randomization, $y_1 \in \mathcal{Y}$ with probability $\alpha$ and $y_2 \in \mathcal{Y}$ with probability $1 - \alpha$. Such binary lotteries have been already examined in an experimental setting, involving human evaluators, by Davidson & Marschak (1958). For these reasons, we name these lotteries Machina-Davidson-Marschak (MDM) Lotteries. Then, instead of solvability, we consider a property comparable to the "Reduction of Compound Lotteries Axiom" (Luce & Raiffa, 1957; Machina & Siniscalchi, 2014), which we call "expandability".

We end up with a simplified treatment of the seminal works of Doignon & Falmagne (1974) and Krantz et al. (1989), yet with MDM's "twist", and use the acronym KLST* to name this framework. We obtain a normative bedrock for choice theory able to connect with Savage *in extenso*, with underlying mechanisms that resonate in the context of RLHF, and most importantly authorizing a natural behavior forbidden in BTL: *abstention*. KLST* is indeed able to encompass the setting where users can decline choosing between $y$ and $y'$ given $x$. Obviously, the triptych (choice, preference, abstention) needs to operate with specific rules to be consistent and useful, and we now describe its three main layers: expandability, local choice structure and monotonicity.

**Expandability** defines the way we include simple, binary choice lotteries. At the human choice's end, stochastic choices on MDM lotteries for lottery $(y_1 y_2)_\alpha$ vs general lottery $L$ are computed as[*]

$$p((y_1 y_2)_\alpha \succ L|x)$$
$$\doteq \quad \alpha \cdot p(y_1 \succ L|x) + (1-\alpha) \cdot p(y_2 \succ L|x). \quad (5)$$

and similarly $p(L \succ (y_1 y_2)_\alpha|x) = \alpha \cdot p(L \succ y_1|x) + (1 - \alpha) \cdot p(L \succ y_2|x)$. Putting a density on $\alpha$ yields a sampling process totally compatible with the triplet sampling process $(x, y, y')$ against which preference optimization is usually done. If deemed preferable or necessary (if e.g. picking $\alpha$ does not follow a stochastic process), one could also consider having the user face lotteries directly in the choices to make. Fix $\alpha \in [0, 1]$. Let $\mathcal{Y}^\alpha \doteq \{(yy')_\alpha : y, y' \in \mathcal{Y}\}$ be the set of available lotteries. An important point to remember is that lottery $(yy)_\alpha$ is equivalent to alternative $y$ because any

---

[*] The general lottery notation $L$ is the same as for a loss function but the objects referred to are clear from context.

choice probability involving $(yy)_\alpha$ breaks down to the one involving only $y$ via (5). Hence $\mathcal{Y}^\alpha$ "contains" $\mathcal{Y}$. We say that choice probabilities are *expandable* to denote that the support of all actions $\mathcal{Y}$ is extended to cover all lotteries.

**Local choice structure** Fix $\alpha \in [0, 1]$. Each $x \in \mathcal{X}$ defines an undirected graphs $G_x^\alpha \doteq (\mathcal{Y}^\alpha, \mathcal{E}_x^\alpha)$ where edges encode *zero abstention* (**ZA**) in a choice between lotteries:

$$(L, L') \in \mathcal{E}_x^\alpha \Leftrightarrow p(L \succ L'|x) + p(L' \succ L|x) = 1. \quad (6)$$

We also introduce the shorthand

$$L \rhd_x L' \quad \Leftrightarrow \quad p(L \succ L'|x) \geq 1/2 \quad (7)$$

($\alpha$ implicit), indicating that lottery $L$ is *preferred* (**P**) over $L'$ given $x$. In an undirected graph, a *wedge* $\{a, b, c\}$ is a sequence of three vertices that form a chain as $(a, b), (b, c)$. Our definition of a local choice structure, which extends to lotteries that of Krantz et al. (1989, Chapter 17), follows.

**Definition 3.2.** Lotteries yield a Local Choice Structure (LCS) iff the three properties are satisfied for any $\alpha \in (0, 1)$:

- **Bearability**: $(L, L) \in \mathcal{E}_x^\alpha, \forall L \in \mathcal{Y}^\alpha$;
- **ZA∧P⇒ZA**: for any wedge $\{L_2, L_1, L_3\}$ in $G_x^\alpha$, if in addition $(L_1 \rhd_x L_2) \wedge (L_1 \rhd_x L_3)$ or $(L_2 \rhd_x L_1) \wedge (L_3 \rhd_x L_1)$ then $(L_2, L_3) \in \mathcal{E}_x^\alpha$;
- **P⇒ZA∧P**: $\forall L, L'$ s. t. $L \rhd_x L'$ (resp. $L' \rhd_x L$), $\exists n \geq 1$ and path $(L = L_0, L_1), (L_1, L_2), ..., (L_{n-1}, L_n = L')$ in $\mathcal{E}_x^\alpha$ s. t. $L_i \rhd_x L_{i+1}, \forall i \in [n-1]$ (resp. $L_{i+1} \rhd_x L_i, \forall i \in [n-1]$).

Bearability is equivalent to asking a user to choose between two copies of the same lottery without abstaining, thus imposing that all lotteries and all alternatives are bearable. The two following axioms are logically named to emphasize interactions between preference (**P**) and zero abstention (**ZA**). The first, **ZA∧P⇒ZA**, says that preference holds insights that alleviate abstention. Suppose $x = S =$ "Indoor activity in $(S)$eoul in July", $L_1 = (L)$ibrary, $L_2 = (Z)$oo, $L_3 = (M)$aze. Now, *suppose* people's profile is such that $Z \rhd_S L$, $M \rhd_S L$ and there is *no* abstention in the related choices, because in the context of the prompt $S$ people would rather do an animal- or exercise-related activity ($Z$ or $M$). Then it makes sense to think that if they were presented with choosing $Z$ vs $M$, they would not abstain in picking one, regardless of the polarity of the choice.

The second, **P⇒ZA∧P**, says that there is always a zero abstention chain "hidden" under a preference: if a lottery is preferred to another, there exists a chain of preferences *without abstention* that connects the two. Suppose $x = K =$ "Looking for ICML keynote speaker" and $L, L'$ two candidates in a poll with a marked preference, say $L \rhd_K L'$, but nonzero abstention because of very different profiles that creates an apples-to-oranges feeling. Then it is reasonable to think that one could extract a list of potential speakers $L_1, L_2, ..., L_{n-1}$ such that people would not abstain in choosing in $\{L_i, L_{i+1}\}$ ($i = 0, 1, ..., n$, $L_0 \doteq L, L_n \doteq L'$):

think of a list of speakers with slowly drifting profiles in the sequence but with clear "orthogonal" comparability (e.g. contract amounts, citations, institutions, collaborations, etc.) so that there would be no will for abstention. Note that the request $\alpha \in (0, 1)$ excludes the extreme choices that would break down lotteries to trivial choices from $\mathcal{Y}$.

**Monotonicity** To wrap up our model, we need a property among *relative preferences* (Krantz et al., 1989).

**Definition 3.3.** Lotteries satisfy monotonicity iff *there exists* $\alpha \in (0, 1)$ such that the following holds. For any $L_1, L_2, L_3, L_4, L_5, L_6$ in $\mathcal{Y}^\alpha$, if (i) $\{L_1, L_2, L_3\}$ is a triangle in $\mathcal{E}_x^\alpha$, (ii) $\{L_4, L_5, L_6\}$ is a wedge in $\mathcal{E}_x^\alpha$, (iii) $p(L_1 \succ L_2|x) \geq p(L_4 \succ L_5|x)$ and (iv) $p(L_2 \succ L_3|x) \geq p(L_5 \succ L_6|x)$, then

$$((L_4, L_6) \in \mathcal{E}_x^\alpha) \wedge (p(L_1 \succ L_3|x) \geq p(L_4 \succ L_6|x)). \quad (8)$$

The large number of parameters may create an impression of complexity which hides a simple core message: condition (iii) may be read "$L_1$ is preferred to $L_2$ **more than** $L_4$ is preferred to $L_5$" and condition (iv) is "$L_2$ is preferred to $L_3$ **more than** $L_5$ is preferred to $L_6$". The right consequence in (8) is then intuitive: "$L_1$ is preferred to $L_3$ more than $L_4$ is preferred to $L_6$". The left consequence in (8), that the wedge on condition (ii) be in fact a triangle in $\mathcal{E}_x^\alpha$, imposes that all six probabilities in the definition be supported by choices without abstention, so it limits uncertainty. Importantly, monotonicity needs to be defined only for one non-trivial $\alpha \in (0, 1)$ (different from the requirements of a LCS).

**KLST\* structure** We now state this key definition.

**Definition 3.4.** The choice probabilities $p(y \succ y'|x)$ have a **KLST\* structure** iff they are expandable and the resulting lotteries (i) yield a local choice structure and (ii) satisfy monotonicity, $\forall x \in \mathcal{X}$.

If we were to impose $G_x^\alpha$ to be a clique, i.e. to never abstain to pick a choice, then this setting could be substantially simplified to the seminal setting of (Debreu, 1958), encompassing as particular case the BTL model. Additional details about KLST\* are in the Appendix, Section I.

## 4. The Expanse

We fully generalize all the analytical choices DPO as in (1), (2), (3) (Section 2.1) using the full power of the normative framework introduced in Section 3.

### 4.1. Step 1, the RLHF objective

Let us generalize (1) to

$$J_{R,r}(\pi(.|x)) \doteq \mathbb{E}_{y \sim \pi(.|x)}[r(x, y)] - R(\pi(.|x) \| \pi_{\text{ref}}(.|x)) \quad (9)$$

with $R : \Delta_n \times \Delta_n \to \mathbb{R}$. What can we require of $R$ ? A first condition makes all the possible choices on the same

minimization scale and require their minimal value to be 0. In ML, a usual constraint is for $R$ to be the *regret* of a function $Q : \Delta_n \times \Delta_n \to \mathbb{R}$, so we are left with eliciting $Q$. A simple constraint for $Q$ is obtained by remarking that RLHF is iterative in nature for policy learning: $\pi_{\text{ref}}$ was first learned, and then (9) is solved. So suppose we have learned

$$\pi_*(.|x) \quad \doteq \quad \arg\max J_{R,r}(\pi(.|x)),$$

and we want to carry out another iteration of training. We must ensure that $\pi_*(.|x)$ is the best initialization, just like $\pi_{\text{ref}}(.|x)$ was for the previous stage, regardless of the rewards to be involved. This leads to a specific request on the objective:

$$\pi(.|x) \quad \in \quad \arg\min_{\pi'(.|x)} Q(\pi(.|x)\|\pi'(.|x)). \quad (10)$$

In the jargon of Savage, this means imposing $Q$ be proper.

**Theorem 4.1.** *If $R$ is the regret of a proper loss then*

$$R(\pi(.|x)\|\pi'(.|x)) \quad = \quad D_{\phi,\boldsymbol{G}}(\pi(.|x)\|\pi'(.|x)) \quad (11)$$

*where $D_{\phi,\boldsymbol{G}}$ is a Bregman divergence. More specifically, we can show $\phi(\boldsymbol{p}) = -L(\boldsymbol{p},\boldsymbol{p})$ for some proper loss $\boldsymbol{\ell}$ (4) and can pick $G_i = -\ell_i$.*

Proof in Appendix, Section II.1. Note that $G_i$ is allowed to depend not just on $y_i$.

### 4.2. Step 2, the link human choice - rewards

Without further ado, we provide the generalization to (2).

**Theorem 4.2.** *Suppose $p(y \succ y'|x)$ has a KLST\* structure. Then there exists*

*1. a strictly increasing function $F$ with $\text{Im}F \subseteq [0,1]$ and such that $F(-z) + F(z) \leq 1, \forall z \in \text{dom}F$;*
*2. a function $u(x,y) : \mathcal{X} \times \mathcal{Y} \to \mathbb{R}$ such that*

$$p(y \succ y'|x) \quad = \quad F(u(x,y) - u(x,y')). \quad (12)$$

Proof in Appendix, Section II.2. Importantly, we now check that the trick allowing DPO to discard the expensive $Z(x)$ normalization from choice probabilities in (2) (Section 2.1) still holds. We get from KKT conditions in the optimization of (9) with $R$ as in (11) the relationship (indexed notations follow Theorem 4.1) for any $i \in [n]$,

$$r(x,y_i) + 0_{\pi(y_i|x)>0} \cdot \mu_{y_i}$$
$$= \quad \lambda + G_i(\pi(.|x)) - G_i(\pi_{\text{ref}}(.|x)), \quad \boldsymbol{G} \in \partial\phi. (13)$$

Here, $\lambda$ is the Lagrange multiplier enforcing normalization of $\pi(.|x)$, $\mu_{y_i} \geq 0$ is the one for the constraint of non-negativity for $\pi(y_i|x)$; finally, for a predicate $\Pi$, $0_\Pi$ takes value 0 if $\Pi$ is true and 1 otherwise. If we plug $u(x,y_i) \doteq$

$r(x,y_i) + 0_{\pi(y_i|x)>0} \cdot \mu_{y_i}$ (if all policy values are $> 0$ such as in DPO, this is just the reward function) in (13), then we remark for any $x,y,y'$,

$$u(x,y_i) - u(x,y_j)$$
$$= G_i(\pi(.|x)) - G_i(\pi_{\text{ref}}(.|x)) - G_j(\pi(.|x)) + G_j(\pi_{\text{ref}}(.|x)) (14)$$

so like in DPO, the expression of $p(y_i \succ y_j|x)$ in Theorem 4.2 does not depend on the (computationally expensive) $\lambda$.

### 4.3. Step 3, the final loss

We finally investigate what eligible $\psi$ can be used in (3) to replace the logistic loss. Minimizing a function $I$ depending on probabilities, Savage's properness framework is again the instrument of choice for the generalization so we learn policy parameters $\boldsymbol{\theta}$ to minimize

$$I(\pi_{\boldsymbol{\theta}}) \quad \doteq \quad \mathbb{E}_{(x,y_i,y_j)\sim\mathcal{D}}\left[\ell(p_{\boldsymbol{\theta}}(y_i \succ y_j|x))\right]. \quad (15)$$

for some *function $\ell : [0,1] \to \mathbb{R}$* which is part of a proper loss definition. Let us progressively connect the dots between $\ell$ and $\psi$ via proper losses.

First, (15) defines a *binary* classification problem where $(x,y_i,y_j)$ is positive if say "$y_i \succ y_j$ given $x$" and negative if "$y_j \succ y_i$ given $x$" (Sun et al., 2025) so the proper loss is scalar valued, which we note $L(p,p) = p\tilde{\ell}_1(p) + (1-p)\tilde{\ell}_0(p)$ after Reid & Williamson (2010). The indexing in $\{0,1\}$ is a convention departing from our indexing in $1, 2, ..., n$ (Definition 3.1): "0" usually refers to the so-called "negative class" and "1" to the "positive class". Such a loss is noted $(\tilde{\ell}_0, \tilde{\ell}_1)$.

Second, the connection naturally follows from the following Theorem, which is the main result of our paper.

**Theorem 4.3.** *For **any** strictly increasing function $\psi : \mathbb{R} \to \mathbb{R}$, **any** strictly increasing function $\tilde{F} : \mathbb{R} \to [0,1]$, there exists a strictly proper loss $(\tilde{\ell}_0, \tilde{\ell}_1)$ such that*

$$\psi(z) \quad = \quad \tilde{\ell}_0 \circ \tilde{F}(z). \quad (16)$$

Proof in Appendix, Section II.3. Crucially for the scope of our paper, the proof is constructive: it gives the expression of the proper loss. Decomposition (16) follows the blueprint of composite losses (Reid & Williamson, 2010). The numerous consequences of Theorem 4.3 are explored in Section 5. Before, let us wrap up with the generalization of (3) that (16) yields via (15). Define for short

$$\Delta_{i,j}(x) \quad \doteq \quad G_i(\pi_{\boldsymbol{\theta}}(.|x)) - G_i(\pi_{\text{ref}}(.|x))$$
$$- G_j(\pi_{\boldsymbol{\theta}}(.|x)) + G_j(\pi_{\text{ref}}(.|x)) \quad (17)$$

(also $= u(x,y_i) - u(x,y_j)$ (14)), for $i,j \in \{1,2,...,n\}, x \in \mathcal{X}$. Then the connection with (15) naturally comes with

$$\ell(p) \quad \doteq \quad \tilde{\ell}_0(1-p), \quad (18)$$
$$p_{\boldsymbol{\theta}}(y_i \succ y_j|x)) \quad = \quad 1 - \tilde{F}(-\Delta_{i,j}(x)). \quad (19)$$

Consider a function $\tilde{F}$ in Theorem 4.3 that satisfies in addition $\tilde{F}(z) + \tilde{F}(-z) \geq 1$. Then function $F(z) \doteq 1 - \tilde{F}(-z)$ is strictly increasing, has $\text{Im}F \subseteq [0, 1]$ and $F(z) + F(-z) = 2 - \tilde{F}(z) - \tilde{F}(-z) \leq 1$ and so (19) can embed a KLST* structure.

## 4.4. Composite, canonical connections and symmetry

Let us summarize first what has been achieved so far: we have created a ML relevant normative bedrock to sustain a generalization of DPO and this generalization retains all of the analytical features that make DPO appealing. This generalization relies on a triptych $(\ell, F, \tilde{\ell})$:

- $\ell : \Delta_n \to \mathbb{R}^n$ defines a proper loss;
- $F : \mathbb{R} \to [0, 1]$ is strictly increasing and satisfies $F(z) + F(-z) \leq 1$;
- $\tilde{\ell} : \Delta_2 \to \mathbb{R}^2$ defines a strictly proper loss.

Theorems 4.1, 4.2 and 4.3 and their related Subsections show how to construct all analytical elements generalizing DPO for any applicable $(\ell, F, \tilde{\ell})$. There is thus considerable freedom in the design of all elements of our generalization of DPO, *and* the complete extent of this freedom is safeguarded by a solid normative bedrock.

**Importantly**, there is absolutely no need for any element in $(\ell, F, \tilde{\ell})$ to be related to the others as they can be totally independent from each other. However, interesting simplifications happen in special cases to which DPO belongs and that we now cover. We say that $\tilde{\ell} \doteq (\tilde{\ell}_0, \tilde{\ell}_1)$ is symmetric iff $\tilde{\ell}_0(p) = \tilde{\ell}_1(1 - p)$ (Nock & Nielsen, 2008).

**Theorem 4.4.** *For any strictly proper loss $(\tilde{\ell}_0, \tilde{\ell}_1)$, denote $\phi(p) \doteq -L(p, p)$ as per (4). Letting*

$$H(p) \doteq \tilde{\ell}_0(p) - \tilde{\ell}_1(p), \tag{20}$$

*we have both $H \in \partial\phi$ and*

$$\phi^\star(z) = \tilde{\ell}_0 \circ H^{-1}(z). \tag{21}$$

*If the loss is also symmetric, then $\tilde{\ell}_1 \circ H^{-1}(z) = \phi^\star(-z)$.*

(the convex conjugate is defined in Subsection 2.2; proof in Appendix, Section II.4). ML's most popular losses for real-valued classification have the expression in (21): logistic, square, Hinge, Matushita, etc. . $\phi$ is strictly convex (Gneiting & Raftery, 2007) and so a canonical link $H$ of the loss, which has a remarkably simple analytical form, is strictly increasing and can be used to craft the human choice's $F$ in the KLST* model. This is stated more precisely now.

**Corollary 4.5.** *For any strictly proper loss $(\tilde{\ell}_0, \tilde{\ell}_1)$, suppose we pick $F = H^{-1}$ in Theorem 4.2 where $H$ is as in (20) and let $\ell \doteq \tilde{\ell}_0$ in (15). Then (15) simplifies to:*

$$I(\pi_{\boldsymbol{\theta}}) = \mathbb{E}_{(x, y_i, y_j)} [\psi(-\Delta_{i,j}(x))]. \tag{22}$$

*with $\psi \doteq \phi^\star$ and $\Delta_{i,j}(x)$ is defined in (17).*

Tying the KLST* model to the loss via the constraint $F = H^{-1}$ is called a canonical connection. Any valid $F$ in $(\ell, F, \tilde{\ell})$ with $\lim_{-\infty} F = 0, \lim_{+\infty} F = 1$ and satisfying additional assumptions is such that $F^{-1} = \tilde{\ell}_0 - \tilde{\ell}_1$ for some strictly proper loss $(\tilde{\ell}_0, \tilde{\ell}_1)$. Using the vocabulary of properness (Reid & Williamson, 2010), this justifies calling the general triptych $(\ell, F, \tilde{\ell})$, without any constraint, a composite connection.

In this very broad picture of the triptych $(\ell, F, \tilde{\ell})$, one sees that DPO occupies a tiny and very specific place: it corresponds to (i) a canonical connection with (ii) a symmetric loss (the log-loss), and also has (iii) the additional constraint $\ell_i = \tilde{\ell}_j, \forall i, j$, as we recall $\tilde{\ell}_0(p) \doteq -\log(1 - p)$ and $\tilde{F}(z) = F(z) = \sigma(z)$.

*Remark 4.6.* If $F = H^{-1}$, condition $F(-z) + F(z) \leq 1$ (Theorem 4.2) is equivalent to requiring $\forall\delta \in (0, 1/2)$,

$$\tilde{\ell}_0\left(\frac{1}{2} + \delta\right) + \tilde{\ell}_0\left(\frac{1}{2} - \delta\right) \geq \tilde{\ell}_1\left(\frac{1}{2} + \delta\right) + \tilde{\ell}_1\left(\frac{1}{2} - \delta\right).$$

While this condition is met for symmetric losses (with equality), if an asymmetric loss $(\tilde{\ell}_0, \tilde{\ell}_1)$ does not meet it, then the "flipped" loss $(\tilde{\ell}_1(1 - p), \tilde{\ell}_0(1 - p))$ does.

## 5. Fallout

In this section, we discuss the repercussions of our theory.

### 5.1. The human choice model vanishes

The reason why Theorem 4.3 is our main result is not just technical: it is also about its consequence in the context of the flurry of refinements or generalizations of DPO.

It is indeed an understatement to say that since DPO, attempts have been flourishing to depart from DPO's choices. However, a huge majority of them still explicitly rely on the BTL part (Alfano et al., 2025; Azar et al., 2024; Chen et al., 2025a; Cui et al., 2024; Gu et al., 2024; Gupta et al., 2025; Hong et al., 2024; Jang et al., 2025; Lee et al., 2025; Li et al., 2025; Lu et al., 2024; Meng et al., 2024; Park et al., 2024; Ramesh et al., 2024; Yuan et al., 2023; Zhao et al., 2025) (and many more). Those trying to depart from BTL either still gravitate very closely to BTL if they remain within a human choice normative model (Chen et al., 2025b; Zhang et al., 2025) or they explicitly abandon any human choice affiliation (Khorasani et al., 2025).

The human choice model, which is such a neat grounded component of DPO with BTL, clearly plays more the role of a straitjacket in follow-ups, either constraining those paths that keep human choice grounding, or leading to risky byways when abandoning any link to human choice.

This straitjacket literally vanishes with Theorem 4.3: **any** applicable $\psi$ defining the key loss in (3) according to Theo-

rem 4.3 can operate with **any** human choice model whose choice probabilities have a KLST* structure. The "hidden" component that makes this alchemy to always work is the loss $\tilde{\ell}$ in the triptych $(\boldsymbol{\ell}, F, \tilde{\boldsymbol{\ell}})$. We insist on the fact that it is not just a "for all applicable $\psi$, there exists a related human choice model such that everything works properly" but really "for all applicable $\psi$ and *for all* related human choice models, everything works properly". Thus, any preference optimization scheme in the DPO filiation can completely forget the human choice part and rely only on two properly defined parameters, which, it turns out, are the key analytical parameters for training:

- The proper loss $\boldsymbol{\ell}$ which defines the reward difference in the final loss (3), (17);
- The function $\psi$ which defines the final loss in (3)

## 5.2. No convex loss no cry

It is not just a question of optimization convenience: Savage's properness framework for real-valued prediction has so far been the ML bedrock of convex losses (e.g. logistic loss) (Nock & Nielsen, 2008; Reid & Williamson, 2010). So it comes as a surprise that Theorem 4.3 only posits strict monotonicity for $\psi$ and does not impose convexity. In fact, a close look at the proof shows that this is a consequence of the interplay between $F$ and $\tilde{\ell}$ in the triptych $(\boldsymbol{\ell}, F, \tilde{\boldsymbol{\ell}})$. Obviously, the way this small analytical knob operates translates in sizable freedom for ML at a time where almost all variations of DPO still rely on a convex choice for $\psi$.

Savage's properness framework also imposes another property to the final loss: it is monotonic (Nock & Nielsen, 2008; Walder & Nock, 2020). It is of independent interest that Theorem 4.3 unlocks non-convexity but still rules out non-monotonic choices for $\psi$, which are in fact core to some DPO extensions (Azar et al., 2024). Interesting but not surprising in our context: non-monotonicity may create misbehaviors for optimization that have been long identified in supervised learning (Hastie et al., 2009, pp 348).

## 5.3. Stranger things

A curious bit of our generalization is that DPO is the **only** choice properly operating in our broad framework given its constraints. What could be seen as a minor remark is in fact crucial in the context of DPO's influence on its followers.

Remind (Subsection 4.4) that DPO has the constraint $\ell_i = \tilde{\ell}_j, \forall i, j$ and so $\boldsymbol{\ell}$ is *separable*: $\ell_i(\boldsymbol{p})$ depends only on $p_i$. The following Theorem has been known and shown under various restrictive flavors for almost seven decades.

**Theorem 5.1.** *Suppose that the loss $L$ in (4) is such that $\boldsymbol{\ell}$ is separable and $L$ is $\boldsymbol{p}$-proper for some target vector $\boldsymbol{p} \in \Delta_n$ with non-zero coordinates at indexes $\mathbb{I} \subseteq \{1, 2, ..., n\}$ with $\mathrm{Card}(\mathbb{I}) > 2$. Then $\ell_i(z) = -K_1 \log(z) + K_2, \forall i \in \mathbb{I}$,*

*where $K_2 \in \mathbb{R}$ and $K_1 > 0$ are constants.*

(Proof in Appendix, Section II.5) This Theorem has crossed almost seven decades under different forms with more restrictive assumption that would have limited its scope in our context: (McCarthy, 1956) mentions an unpublished result imposing all $\ell_i$s to be the same; the proof of (Savage, 1971, Section 9.4) imposes strict properness and differentiability on $\boldsymbol{\ell}$, and show the result for vectors $\boldsymbol{p}$ partially uniform; (Bernardo, 1979, Theorem 2) imposes smoothness, full support and all $\ell_i$s to be the same; Dawid (2007) imposes differentiability and all $\ell_i$s to be the same. While it has the benefit of covering *all* proper losses, our generalization is formulated to insist on the fact that there is no "lucky" target $\boldsymbol{p}$ for which the loss in (10) could be different from the KL divergence[†]. The Theorem also has maximal generality (it breaks when $\mathrm{Card}(\mathbb{I}) = 2$, see e.g. Table 1).

Theorem 5.1 is important because it shows that departing from KL imposes departing from DPO's blueprint for losses. This has little to zero impact on the papers essentially keeping the KL divergence in Step 1 (Subsection 4.1) or DPO's log-ratio policy dependence (Azar et al., 2024; Chen et al., 2024; Sun et al., 2025; Ethayarajh et al., 2024; Meng et al., 2024; Zhao et al., 2023; Xu et al., 2024a; Zhao et al., 2025; Slocum et al., 2025; Shao et al., 2025; Choi et al., 2025; Yang et al., 2025; Xiao et al., 2025). However, for others clearly wanting to depart from KL, the risk is real to break properness (Alfano et al., 2025; Gupta et al., 2025). Separability may have the computational advantage that each coordinate of $\boldsymbol{\ell}$ depends only on one action ($n$ can be huge). Fortunately, there is a simple way to depart from KL, keep properness and retain this advantage. It exploits a simple trick well known in supervised learning.

**Lemma 5.2.** *For any loss $(\ell_0, \ell_1)$, define $\boldsymbol{\ell}^b : \Delta_n \to \mathbb{R}^n$ as $\ell_i^b(\boldsymbol{p}) \doteq \ell_0(p_i) + \sum_{j \neq i} \ell_1 (1 - p_j)$. If $(\ell_0, \ell_1)$ is (strictly) proper, then $\boldsymbol{\ell}^b$ is (strictly) proper.*

The proof, straightforward, is provided for completeness in Section II.6. It shows $L(\boldsymbol{p}, \boldsymbol{p}) = \sum_i p_i \ell_0(p_i) + (1 - p_i)\ell_1 (1 - p_i)$, which indeed keeps the advantage set above. ORPO uses this design (Hong et al., 2024).

## 5.4. Beyond the DPO-verse

Quite surprisingly perhaps, some approaches that have added new layers to DPO can also be supported by our normative framework *as is*: home advantages, margins and length normalization.

**Home advantages / margins** have been introduced by Meng et al. (2024), whereby function $\psi(z)$ in (3) includes a slack in its argument, $\psi(z - \gamma)$, transposing to DPO the ad-hoc "home advantage" introduced in data analysis (Agresti,

---

[†]Clearly, $\mathrm{Card}(\mathbb{I}) > 2$ in RL, even for the simplest of toy problems (Xu et al., 2024b).

| Loss | $\ell_i(\boldsymbol{p})$ | proper if $n=2$? | proper if $n>2$? | $F(z)$ | $\phi^\star(-z)$ | |
|---|---|---|---|---|---|---|
| log | $-\log(p_i)$ | ✓ | ✓ | $1/(1+\exp(-z))$ | $\log(1+\exp(-z))$ | (Rafailov et al., 2023) |
| binary entropy | $-\log(p_i)+\sum_{j\neq i}-\log(1-p_j)$ | ✓ | ✓ | $1/(1+\exp(-z/2))$ | $2\log(1+\exp(-z/2))$ | (Hong et al., 2024) |
| square ($\tau>0$) | $\frac{1}{\tau}\cdot(1-p_i)^2$ | ✓ | ✗ | $\max\left\{0,\min\left\{\frac{1+\tau z}{2},1\right\}\right\}$ | $\tau\cdot\begin{cases}\frac{1}{4\tau}-\frac{x}{\tau} & \text{if } x<0 \\ \left(x-\frac{1}{2\tau}\right)^2 & \text{if } 0\leq x\leq\frac{1}{2\tau} \\ 0 & \text{if } x>\frac{1}{2\tau}\end{cases}$ | (Azar et al., 2024) |
| Matushita ($\mu\geq0$) | $\mu\cdot\sqrt{\frac{1-p_i}{p_i}}$ | ✓ | ✗ | $\frac{1}{2}\cdot\left(1+\frac{x}{\sqrt{x^2+\mu^2}}\right)$ | $\frac{-z+\sqrt{\mu^2+z^2}}{2}$ | (Yuan et al., 2023) |
| alpha ($\beta\geq0$) | $(1-p_i^{-\beta})/\beta$ | ✓/✗ | ✓/✗ | No general closed form | | (Gupta et al., 2025) |

*Table 1.* DPO variants in the PPPO framework (see text). Each loss may be used either as $\boldsymbol{\ell}^a$ or $\boldsymbol{\ell}^b$ in PPPO. The properness "regime" is indicated for each loss, either for $n=2$ (typically for $(\ell_0^a,\ell_1^a)$ in PPPO) or $n>2$ (for $\boldsymbol{\ell}^b$ in PPPO).

2012, pp 438), with $\gamma$ a so-called margin term. There is no formal justification for those, but it turns out that this "slack" is authorized by the following simple Lemma.

**Lemma 5.3.** *Let $(\ell_0,\ell_1)$ be proper (resp. strictly proper), and $a>0$ and $c\in\mathbb{R}$ any constants. Then the loss $(\tilde{\ell}_0,\tilde{\ell}_1)\doteq(a\ell_0,a\ell_1+c)$ is proper (resp. strictly proper) and furthermore, using notations from Theorem 4.4 we have the following key quantities for loss $(\tilde{\ell}_0,\tilde{\ell}_1)$: $\tilde{H}^{-1}(z)=H^{-1}((z+c)/a)$ and $\tilde{\phi}^\star(z)=a\phi^\star((z+c)/a)$.*

(Proof in Appendix, Section II.7) Hence the loss setting of (Meng et al., 2024) falls into PPPO as long as $c/a$, which is the $\gamma$ term in SimPO, is non-negative for the stochastic choice model to be KLST* compliant. This is the case in all the papers we have spotted using the margin term (Meng et al., 2024; Zhao et al., 2025). This can also support to some extent techniques that would *tune* the slack (Cui et al., 2024), i.e. replacing constants $a,c$ by functions.

**Length normalization** has been introduced in several ways[‡] to express the fact that $y_i$ is the "level" at which preferences are formulated but $\pi(y_i|.)$ is a compound probability aggregating those of "elementary" actions $y_{i,k},k=1,2,...,n_i$ — tokens in the jargon of LLMs (Meng et al., 2024; Gu et al., 2024; Park et al., 2024; Li et al., 2025). No such dependence is reflected in any of the three steps in Subsections 4.1, 4.2 and 4.3. We show how the framework of Subsection 4.1 also supports the popular corrections in Meng et al. (2024); Gu et al. (2024); Park et al. (2024); Li et al. (2025). We first observe that for autoregressive models, we have

$$\pi(y_i|x)=\prod_{k=1}^{n_i}\pi(y_{i,k}|y_{i,<k},x),\forall i=1,2,...,n_i. \quad (23)$$

Suppose we wish to compute some $\tilde{\pi}(y_i|x)$ that would satisfy two constraints: (a) it is the *best probability approximation* to all $\{\pi(y_{i,k}|y_{i,<k},x)\}_{i=1}^{n_i}$, $\tilde{\pi}(y_i|x)$ and (b) it has the form $\tilde{\pi}(y_i|x)=u(\pi(y_i|x))$ via (23), for some $u$ to specify that would carry some knowledge about $\{\pi(y_{i,k}|y_{i,<k},x)\}_{i=1}^{n_i}$ – the simplest of which being the size, $n_i$. Plugging $\tilde{\pi}(y_i|x)=u(\pi(y_i|x))$ in (14) would then carry this knowledge at zero cost in the DPO pipeline. Formulating (a) is just a matter of reusing (9) by using

---

[‡]The usual ad-hoc justification is reward modification (Park et al., 2024).

$\pi(y_{..}|y_{..},x)$ as reference policy $\pi_{\text{ref}}$, assuming all rewards equal — and of course after Theorem 4.1, using a Bregman divergence $D_{\phi,\boldsymbol{G}}$ as criterion so we end up looking at the argument probability maximizing (for all $i=1,2,...,n$)

$$J_\phi(\tilde{\pi}(y_i|x))$$
$$\doteq -\sum_{k=1}^{n_i}D_{\phi,\boldsymbol{G}}(\tilde{\pi}(y_i|x)\|\pi(y_{i,k}|y_{i,<k},x)). \quad (24)$$

**Lemma 5.4.** *Take $\phi(\boldsymbol{p})\doteq\sum_i p_i\log p_i$ in (24) (KL divergence). Then the solution to (24) satisfies*

$$\log\tilde{\pi}(y_i|x) = (1/n_i)\cdot\log\pi(y_i|x),$$

*which is the approach of Meng et al. (2024). If instead one picks $\phi(\boldsymbol{p})\doteq\sum_i-\log p_i$ (Itakura-Saito divergence) then the solution to (24) satisfies this time*

$$\forall i,\exists\alpha_i>0:\log\tilde{\pi}(y_i|x)=\log\pi(y_i|x)+\alpha_i n_i, \quad (25)$$

*i.e. the approach of Park et al. (2024); Chen et al. (2025a).*

The proof, Appendix Section II.8, makes $\alpha_i$ in (25) more precise and suggests an eventual bias in the formula, with a simple suggestion to debias it.

### 5.5. Across the KLST*-verse

Our human choice model allows for abstention. In human choice theory or applied econometrics, the usual approach is instead to introduce an "outside option" (as a new element in $\mathcal{Y}$), which allows for the derivation of a reservation utility/reward (Train, 2009, Chap. 2). However, the context in which we operate is different: No-choice between two options reflects a local incomparability rather than global rejection (where neither option reaches the reservation utility threshold). The advantage of our local structure is analytical: rather than complicating the framework with a third global and intransitive utility, it "absorbs" abstention behavior directly into the choice probabilities.

BTL is closely linked to Random Utility Theory (RUT), according to which the utility of an option is a random variable, composed of a deterministic component and a

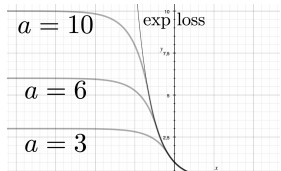

| $a$ | win% (us) |
|---|---|
| 10 | 44.60% |
| 6 | **54.50%** |
| 3 | **53.00%** |

*Figure 1.* Left: $\psi_a$ (26) for the three values of $a$ considered against exponential loss.; Right: comparison of training with $\psi_a$ vs exponential loss. A value >50% means $\psi_a$ wins (see text).

stochastic error term. When these random variables are assumed to be independent and follow a specific distribution (i.e. Gumbel), the RUT framework naturally leads to BTL. Our axiomatic framework also yields to a monotonic utility function, but without relying on explicit error terms. A key advantage is that a utility representation is obtained without being constrained by the strict symmetry of sum-equality-to-one imposed by standard RUT models with Gumbel noise.

### 5.6. It's a small world (but it doesn't have to be)

We make more concrete the fact that many variations of DPO are in fact very close to DPO at the scale of the "territory" that we cover.

**Definition 5.5.** For any losses $(\ell_0^a, \ell_1^a)$ (strictly proper) and $\ell^b$ (proper), the PPPO (Proper-Proper Preference Optimization) framework is the setting of the following parameters in Theorems 4.1, 4.2 and 4.4: (i) $\ell \doteq \ell^b$ in Theorem 4.1, (ii) $F \doteq (\ell_0^a(p) - \ell_1^a(p))^{-1}$ in Theorem 4.2, (iii) $(\ell_0, \ell_1) \doteq (\ell_0^a, \ell_1^a)$ in Theorem 4.4.

PPPO is the canonical connection studied in Subsection 4.4. Table 1 displays that prominent examples of DPO variations belong to this PPPO setting. RRHF keeps $\ell^b$ from DPO but uses Matushita for $\ell^a$ ($\lim_{\mu \to 0} \phi^\star(-z) = \max\{0, -z\}$). ORPO (Hong et al., 2024) is an interesting case which keeps $\ell^a$ from DPO but uses the binary entropy for $\ell^b$. IPO (Azar et al., 2024) is also interesting because, while it keeps $\ell^b$ from DPO but picks the square loss for $\ell^a$, scraps the asymptotes from the convex surrogate. Finally, AlphaPO is also interesting as it keeps DPO's $\ell^a$ but replaces the log-loss in $\ell^b$ by a tempered variant (Nock et al., 2023) which can break properness (Sypherd et al., 2022).

## 6. Toy experiment

Most of the map that we uncover is still blank in terms of SOTA, and it cannot be the purpose of this (or even just one) paper to show how to "navigate" it experimentally. However, as a toy illustration, we dig at one example "outlier" location. Consider the following loss:

$$\psi_a(z)$$
$$\doteq (z < \log(2/a)) \; ? \; a - (a^2/4) \cdot \exp(z) : \exp(-z) \quad (26)$$

$\lim_{a \to -\infty} \psi_a(z) = \exp(-z)$ so $\psi_a$ can encode the exponential loss. Otherwise, $\psi_a$ is non-convex and balances the strong convexity of the exponential loss ($a = -\infty$) with Lipschitzness for $a \neq -\infty$. We use model gemma2_2b_it and compare three finite values of $a$ to the exponential loss. Figure 1 presents results obtained for three values of $a$, along with losses' plots (tests on Alpaca Eval v2; rater = gemini-2.5-flash-lite; full details and bigger plots in Appendix, Section III), averaged over two runs. We observe that non-convexity can bring a leverage, which encourages a broader experimental coverage of the topic.

## 7. Conclusion: to be normative (or not to be)

We show that DPO is a particular point in a huge panorama of losses and preference models, that everywhere keeps all of DPO's neat design tricks and properties and complies *in extenso* with a solid normative bedrock. There is thus a large unexplored map to explore for substantial variations to the specific DPO choices, and this map can accommodate a number of additional requirements that have been formulated in the realm of LLMs (it naturally embeds home advantages, margins and length correction; it is not affected by "orthogonal" extensions to DPO that would deal with marginal/posterior optimization, noise handling, time-varying preferences, etc., that can be used without change). Given the variety of environments, requirements, data specificities of LLM training, development, and deployment, it is important to frame the full freedom offered, to then be able to make choices without losing any of its advantages. Such freedom is mandatory to be able escape eventual pitfalls of specific choices (Khorasani et al., 2025).

Normative frameworks have been crucial to safeguard broad pans of ML (supervised learning, Reid & Williamson (2010; 2011), unsupervised learning, Banerjee et al. (2005); Tenenbaum et al. (2000), etc.). Also, ML-at-large is facing renewed scrutiny (Schaeffer et al., 2025) but playing a race against time: our paper is the first to unlock the full potential of DPO's normative framework, yet alternatives to DPO's normative framework have been proposed *even before* such a full understanding was available (Zhi-Xuan et al., 2024).

Yet a normative theory can only deliver on its premises and it would be quite impossible to extend our results using Savage's theory. What this means is that some DPO variants definitely escape our framework (Huang et al., 2025; Wang et al., 2024) – even when they can be very close[§] – and signals an interesting open problem.

---

[§]For example, since $\log(z) = \log(z-1+1) \sim_1 (z-1)-(z-1)^2/2 + o((z-1)^2)$, we derive that $\chi^2$ divergence is a 2nd-order approximation of KL divergence (Huang et al., 2025).

## Acknowledgments

The authors warmly thank Robert C. Williamson and the reviewers (especially reviewer T1bk) for many comprehensive, technical comments and discussions about this material that helped to improve its content.

## Impact statement

This paper presents theoretical work whose goal is to advance the field of machine learning. A priori, we feel that our main result – that preference optimization is disentangled in human choice – may have a positive impact by reducing the risk of bias that can come from picking a specific *model* of human choice to craft a DPO-style algorithm. As we get "rid of" this need, the focus for potential impact shifts from being analytical (e.g. "what is the impact of the function replacing DPO's sigmoid ?") to being normative: what is the impact of the assumptions' bedrock upon which our work relies ? Savage's properness has been known, adopted and used in ML for decades. We feel that interesting lessons can come from the assumptions defining a KLST* structure, especially when it comes to fairness and biases for sensitive domains, so our theoretical work could have a concrete impact on experimental designs touching on human behaviour. But this is beyond the scope of our paper.

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

# Appendix

To differentiate with the numberings in the main file, the numbering of Theorems, etc. is letter-based (A, B, ...).

## Table of contents

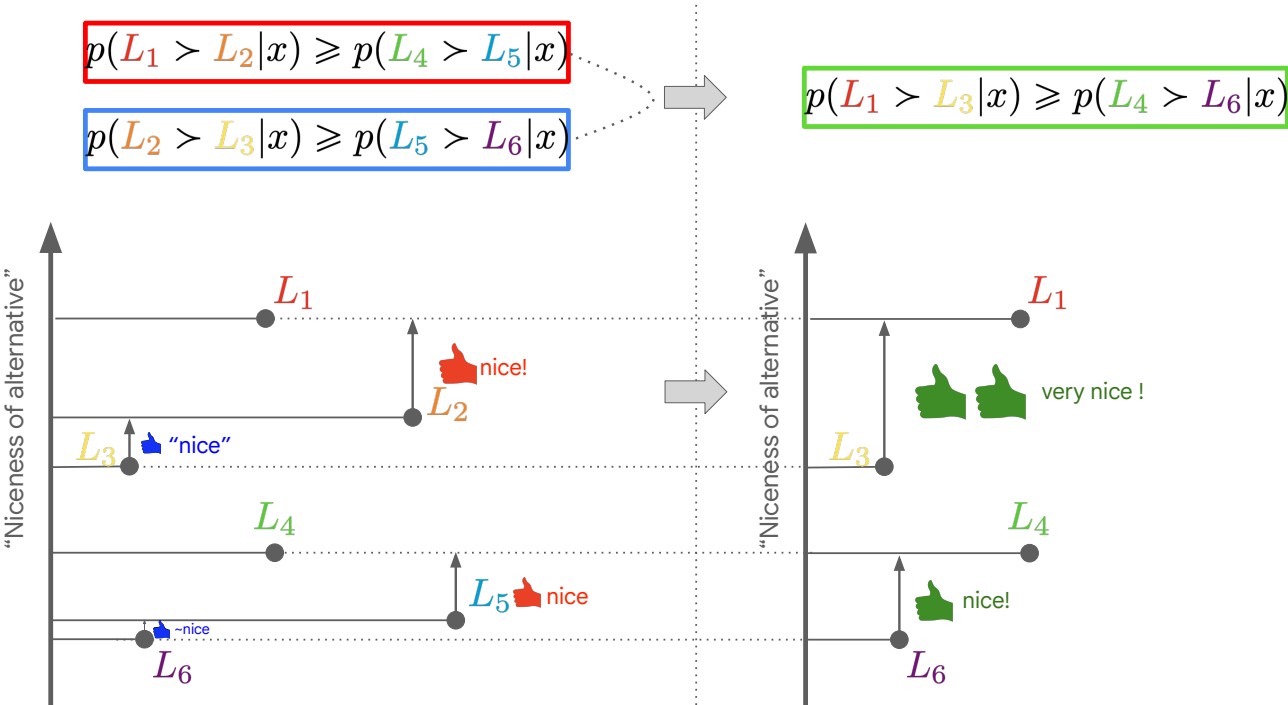

*Figure 2.* The monotonicity property implicitly creates a real valuation of the "niceness" of alternatives in $\mathcal{Y}^{\alpha}$. For example, computing $p(L_1 \succ L_2|x)$ amounts to making a difference between the mappings of $L_1$ and $L_2$ (left, in red). The inequality in (iii), shown in the red rectangle, establishes an order between the related differences along the axis (left), and similarly for the blue rectangle. The mapping then authorizes to compare new related differences, hence probabilities, and derive the right proposition in (8) (main file).

# I. Supplementary material on KLST*

## I.1. Illustration of the Monotonicity Assumption

Figure 2 shows what monotonicity creates "behind the curtain" a mapping for the elements in $\mathcal{Y}^{\alpha}$ on a real axis, representing their "niceness" and allowing for a simple comparison of choice probabilities. We have adopted the same polarity for the difference of the valuations (all arrows go up), but the picture would lead to the same conclusion had we flipped some of them. Indeed, denote for short $z_1, z_2, ..., z_6$ the "hypothetical" coordinates of the lotteries. Then the antecedent of monotonicity on choice probabilities (Figure 2) also read:

$$
\begin{aligned}
z_1 - z_2 &\geq z_4 - z_5, \\
z_2 - z_3 &\geq z_5 - z_6.
\end{aligned}
$$

and simple algebra yields after summing those:

$$
z_1 - z_3 \geq z_4 - z_6,
$$

which is the consequent of the monotonicity condition on choice probabilities.

## I.2. Illustration of Assumption ZA∧P⇒ZA

Figure 3 presents a simple illustration for assumption **ZA∧P⇒ZA**: an edge, e.g. $(L_1, L_2)$, indicates no abstention among lotteries $L_1$ and $L_2$, so one must be preferred to the other one ("$\rhd_x$"). In a wedge (left of Figure 3), if it happens that the central node (here, $L_1$) is preferred to both other lotteries – or both other lotteries are preferred to it –, then it is reasonable to think that the rational mechanisms entailing those preferences without abstention would at least prevent abstention if / when choosing between the wedge's extreme nodes (here, $L_2$ and $L_3$), so $(L_2, L_3) \in \mathcal{E}_x^{\alpha}$. However, it does not necessarily allow to find out which of the two lotteries would be preferred to the other one, hence there is no assumption on the polarity of the preference relationship between the two lotteries.

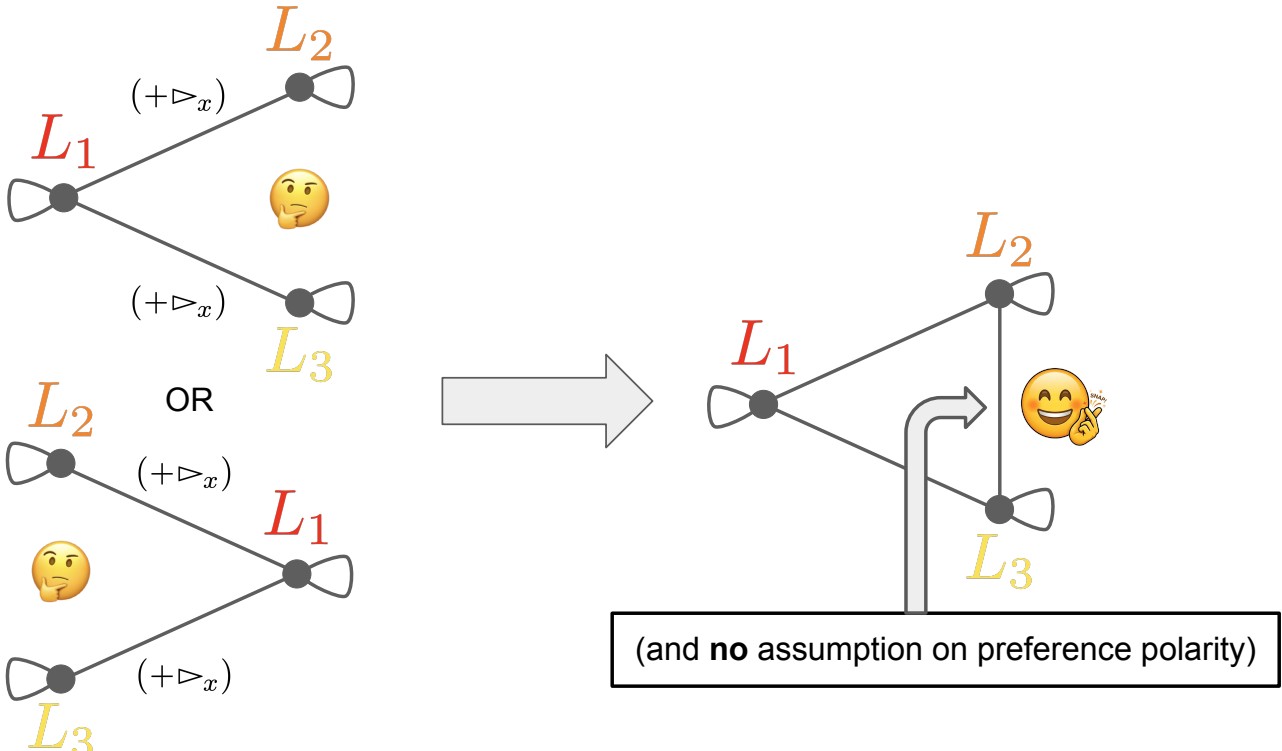

*Figure 3.* An illustration of assumption **ZA**∧**P**⇒**ZA** in graph $G_x^\alpha$ (loops indicate bearability assumption). On the left part, the thinking emoji indicates eventual abstentions in the choice $L_2$ vs $L_3$, which are in fact resolved if it happens that the polarity of preferences in the wedge with respect to the central node of the wedge are the same (right part of the figure). See text for details.

## II. Supplementary material on proofs

### II.1. Proof of Theorem 4.1

Williamson et al. (2016, Proposition 7) shows the standard connection: $Q$ being proper, it necessarily satisfies

$$Q(\pi(.|x)\|\pi'(.|x)) \quad = \quad K(\pi(.|x)) + D_{\phi,G}(\pi(.|x)\|\pi'(.|x)), \tag{27}$$

where $D_{\phi,G}$ is a Bregman divergence. They also show that $K(\pi(.|x)) = Q(\pi(.|x)\|\pi(.|x))$, which leads to the expression of the regret, satisfying by definition:

$$R(\pi(.|x)\|\pi'(.|x)) \quad \doteq \quad Q(\pi(.|x)\|\pi'(.|x)) - Q(\pi(.|x)\|\pi(.|x)).$$

However, we need to show one additional bit to fully prove Theorem 4.1, namely that $-\boldsymbol{\ell} \in \partial - L$, which is simple but yet needs to be shown for completeness. Properness condition imposes $\boldsymbol{p}^\top \boldsymbol{\ell}(\boldsymbol{p}) \leq \boldsymbol{p}^\top \boldsymbol{\ell}(\boldsymbol{q})$ for all $\boldsymbol{p}, \boldsymbol{q} \in \Delta_n$. Put all on the right hand side and add there $-L(\boldsymbol{q}, \boldsymbol{q}) + L(\boldsymbol{q}, \boldsymbol{q})$. Expand with the definition of $L$ and it yields:

$$-\boldsymbol{p}^\top \boldsymbol{\ell}(\boldsymbol{p}) - (-\boldsymbol{q}^\top \boldsymbol{\ell}(\boldsymbol{q})) - (\boldsymbol{p} - \boldsymbol{q})^\top \boldsymbol{\ell}(\boldsymbol{q}) \quad \geq \quad 0,$$

which implies indeed $\boldsymbol{\ell} \in \partial - L$. The convexity of $-L(\boldsymbol{q}, \boldsymbol{q})$ was originally shown in Gneiting & Raftery (2007).

### II.2. Proof of Theorem 4.2

We exploit a classical result of Krantz et al. (1989, Chapter 17, Theorem 2), in two steps: we first show that the local choice structure and monotonicity of lotteries extend to alternatives of $\mathcal{Y}$ (i.e. for $\alpha = \{0, 1\}$) and then we show that considering lotteries allows to bypass the use of a quite non intuitive *solvability* axiom (we separately note that it also appears in Debreu (1958)'s theory, thus when zero abstention is enforced).

Let us start by step one. We cover the properties involved in the local choice structure and monotonicity for lotteries and show that they **specialize** to the lotteries atomic components, i.e. they hold for $\mathcal{Y}$ as well. The following simple Lemma

is central to our results. It actually contains more results than strictly needed for our purpose as those shall be used in a discussion comparing the original KLST model to our lottery model. One can infer from the definition of zero abstention that it is a stronger notion than preference (zero abstention implies a preference from its definition (6), but the converse is not necessarily true). The Lemma shows such a property holds at the "deeper" stage of alternatives.

**Lemma A.** $\forall y_1, y_2, y_3, y_4 \in \mathcal{Y}$, *the following holds*

*(I) on preferences (**P**):*

$$\forall \alpha \in (0,1) : (y_1 y_2)_\alpha \rhd_x (y_3 y_4)_\alpha \quad \Rightarrow \quad (y_1 \rhd_x y_3) \wedge (y_2 \rhd_x y_4), \tag{28}$$

$$\exists \alpha \in (0,1) : (y_1 y_2)_\alpha \rhd_x (y_3 y_4)_\alpha \quad \Leftrightarrow \quad (y_1 \rhd_x y_3) \vee (y_1 \rhd_x y_4) \vee (y_2 \rhd_x y_3) \vee (y_2 \rhd_x y_4). \tag{29}$$

*Furthermore, if among the four alternatives / actions in (29), three are identical and **Bearability** holds, then the equivalence holds with $\rhd_x$ between the two distinct elements on the right proposition: for example, if $y \doteq y_1 = y_2 = y_3 \neq y_4$, then the RHS is $y \rhd_x y_4$;*

*(II) on zero abstention (**ZA**):*

$$\exists \alpha \in (0,1) : ((y_1 y_2)_\alpha, (y_3 y_4)_\alpha) \in \mathcal{E}_x^\alpha \quad \Leftrightarrow \quad ((y_1, y_3) \in \mathcal{E}_x) \wedge ((y_1, y_4) \in \mathcal{E}_x)$$
$$\wedge ((y_2, y_3) \in \mathcal{E}_x) \wedge ((y_2, y_4) \in \mathcal{E}_x). \tag{30}$$

*Proof.* To prove (28), we decompose using (5)

$$p((y_1 y_2)_\alpha \succ (y_3 y_4)_\alpha | x) \quad = \quad \alpha^2 p(y_1 \succ y_3 | x) + \alpha(1-\alpha) p(y_1 \succ y_4 | x)$$
$$+ \alpha(1-\alpha) p(y_2 \succ y_3 | x) + (1-\alpha)^2 p(y_2 \succ y_4 | x), \tag{31}$$

and we have $p((y_1 y_2)_\alpha \succ (y_3 y_4)_\alpha | x) \geq 1/2$. If $p(y_1 \succ y_3 | x) < 1/2$, then there are values of $\alpha$ close to 1 such that $p((y_1 y_2)_\alpha \succ (y_3 y_4)_\alpha | x) < 1/2$, and that is the same if $p(y_2 \succ y_4 | x) < 1/2$ for values of $\alpha$ close to 0. Hence, we must have $y_1 \rhd_x y_3$ and $y_2 \rhd_x y_4$. This is the most we can get because if $p(y_1 \succ y_3 | x) = p(y_2 \succ y_4 | x) = 1$, then we already have $p((y_1 y_2)_\alpha \succ (y_3 y_4)_\alpha | x) \geq \alpha^2 + (1-\alpha)^2 \geq 1/2$ so we cannot enforce either $y_1 \rhd_x y_4$ or $y_2 \rhd_x y_3$.

To prove (29), we just have to remark that $\alpha^2 + 2\alpha(1-\alpha) + (1-\alpha)^2 = 1$. We then prove (30) and note that $1 = p((y_1 y_2)_\alpha \succ (y_3 y_4)_\alpha | x) + p((y_3 y_4)_\alpha \succ (y_1 y_2)_\alpha | x)$ is equivalent to, after decomposition using (5),

$$1 \quad = \quad \alpha^2 (p(y_1 \succ y_3 | x) + p(y_3 \succ y_1 | x)) + \alpha(1-\alpha)(p(y_1 \succ y_4 | x) + p(y_4 \succ y_1 | x))$$
$$+ \alpha(1-\alpha)(p(y_2 \succ y_3 | x) + p(y_3 \succ y_2 | x)) + (1-\alpha)^2 (p(y_2 \succ y_4 | x) + p(y_4 \succ y_2 | x)),$$

which, since $\alpha^2 + 2\alpha(1-\alpha) + (1-\alpha)^2 = 1$, leads to the statement. We tackle the last statement, if among the four elements in (29), three are identical. We see from (31) two probabilities are identical (say $p$) and two are $1/2$ from assumption Bearability; furthermore the coefficient of the probability $p$ is $\alpha$ and that of $1/2$ is $1 - \alpha$, or we permute those, which, if we look at inequality $p((y_1 y_2)_\alpha \succ (y_3 y_4)_\alpha | x) \geq 1/2$, ends up simplifying the inequality as $p \geq 1/2$. $\square$

**Specialization of Bearability** It follows by considering all lotteries of the form $L \doteq (yy)_\alpha$ for $y \in \mathcal{Y}$: $(L, L) \in \mathcal{E}_x^\alpha$ implies after using the property of lotteries (5) $1 \doteq p(L \succ L | x) + p(L \succ L | x) = (\alpha^2 + 2\alpha(1-\alpha) + (1-\alpha)^2) \cdot (p(y \succ y | x) + p(y \succ y | x)) = p(y \succ y | x) + p(y \succ y | x)$, hence $(y, y) \in \mathcal{E}_x$ and bearability holds for $G_x$. Hence, Bearability holds on alternatives / actions as well.

**Specialization of ZA∧P⇒ZA** Take $y_1, y_2, y_3 \in \mathcal{Y}$, with (i) $\{y_2, y_1, y_3\}$ a wedge in $\mathcal{E}_x$ and (ii) $(y_1 \rhd_x y_2) \wedge (y_1 \rhd_x y_3)$. Create three lotteries for any fixed $\alpha \in (0,1)$: $L_1 \doteq (y_1 y_1)_\alpha$, $L_2 \doteq (y_1 y_2)_\alpha$ and $L_3 \doteq (y_1 y_3)_\alpha$. Point (II) in Lemma A and (i) yield immediately that $L_2, L_1, L_3$ is a wedge in $\mathcal{E}_x^\alpha$. We also remark

$$p(L_1 \succ L_2 | x) \quad \doteq \quad p((y_1 y_1)_\alpha \succ (y_1 y_2)_\alpha | x),$$

so the last point of Lemma and the fact that $y_1 \rhd_x y_2$ imply $L_1 \rhd_x L_2$. Similarly, replacing $L_2$ by $L_3$ yields $L_1 \rhd_x L_3$, and we conclude from assumption ZA∧P⇒ZA on lotteries that $(L_2, L_3) \in \mathcal{E}_x^\alpha$, which, from the first point in Lemma A, yields $(y_2, y_3) \in \mathcal{E}_x$. We would have arrived at the same conclusion if we had replaced (ii) by $(y_2 \rhd_x y_1) \wedge (y_3 \rhd_x y_1)$. Hence, ZA∧P⇒ZA holds on on alternatives / actions as well.

**Specialization of P⇒ZA∧P**   Take any $y, y' \in \mathcal{Y}$ such that $y \rhd_x y'$. It comes $L \rhd_x L'$ for $L \doteq (yy)_\alpha$ and $L' \doteq (y'y')_\alpha$. We show the result by induction on $n$. If $n = 1$, then we get $L \rhd_x L'$ (which we knew) and $(L, L') \in \mathcal{E}_x^\alpha$, which implies $(y, y') \in \mathcal{E}_x$ from point (II) of Lemma A, and concludes the specialization proof for $n = 1$. Suppose the property true for any $n \geq 1$ and we show it for $n + 1$. Consider the path $((yy)_\alpha = L_0, L_1), (L_1, L_2), ..., (L_{n-1}, L_n), (L_n, L_{n+1} = (y'y')_\alpha)$ in $\mathcal{E}_x^\alpha$ and such that $L_i \rhd_x L_{i+1}, \forall i \in [n]$. Denote for short $L_i \doteq (y_{i_1}, y_{i_2})$ ($y_{0_1} = y_{0_2} = y$ with a slight abuse of notation). We now exploit the induction hypothesis on the red part of the path: point (I) in Lemma A yields

$$(y_{i_1} \rhd_x y_{(i+1)_1}) \wedge (y_{i_2} \rhd_x y_{(i+1)_2}), \forall i = 0, 1, ..., n-1, \tag{32}$$

And point (II) in Lemma A yields

$$((y_{i_1}, y_{(i+1)_1}) \in \mathcal{E}_x) \wedge ((y_{i_2}, y_{(i+1)_2}) \in \mathcal{E}_x), \forall i = 0, 1, ..., n-1. \tag{33}$$

The last element of the path, $(L_n = (y_{n_1} y_{n_2})_\alpha, L_{n+1} = (y'y')_\alpha) \in \mathcal{E}_x^\alpha$ with $(y_{n_1} y_{n_2})_\alpha \rhd_x (y'y')_\alpha$ yields respectively $((y_{n_1}, y') \in \mathcal{E}_x) \wedge ((y_{n_2}, y') \in \mathcal{E}_x)$ (point (II) Lemma A) and $(y_{n_1} \rhd_x y') \wedge (y_{n_2} \rhd_x y')$ (point (I) Lemma A), which, along with (32) and (33), allows to conclude on the specialization of P⇒ZA∧P. Hence, P⇒ZA∧P holds on on alternatives / actions as well.

**Specialization of Monotonicity**   This one is even simpler: suppose monotonicity holds for **some** $\alpha \in (0, 1)$ and for any $y_1, y_2, y_3, y_4, y_5, y_6$ in $\mathcal{Y}$, build lotteries $L_1 \doteq (y_1, y_1)_\alpha$, $L_2 \doteq (y_2, y_2)_\alpha$, and so on until $L_6 \doteq (y_6, y_6)_\alpha$. Using the fact that, by definition and for example

$$p(L_1 \succ L_2 | x) = (\alpha^2 + 2\alpha(1-\alpha) + (1-\alpha)^2) \cdot p(y_1 \succ y_2 | x) = p(y_1 \succ y_2 | x),$$

we easily get (iii) and (iv) valid over the corresponding alternatives / actions as well. Point (ii) in Lemma A allows to conclude that points (i) (triangle) and (ii) (wedge) of monotonicity also transfers on to alternatives / actions.

At this point, we have completed the first step of our proof. We go on to the second step, showing first that expandability implies the following result on $\mathcal{Y}$: suppose $p_1 \doteq p(y_1 \succ y | x) < p(y_2 \succ y | x) \doteq p_2$. Then for any $q \in (p_1, p_2)$, if we fix

$$\alpha = \frac{q - p_1}{p_2 - p_1} \quad (\alpha \in (0, 1)), \tag{34}$$

then the lottery $(y_2 y_1)_\alpha$ satisfies $p((y_2 y_1)_\alpha \succ y | x) = q$ from (5). This shows that Krantz et al. (1989, Chapter 17, Theorem 2)'s solvability axiom (3) is satisfied. Our definition of local choice structure implies their axioms (1) and (4) and their monotonicity axiom is ours and so their Theorem 2 implies the existence of $u(x, y) : \mathcal{X} \times \mathcal{Y} \to \mathbb{R}$ such that for any $y_1, y_2, y_3, y_4 \in \mathcal{Y}$ and $x \in \mathcal{X}$,

$$p(y_1 \succ y_2 | x) \leq p(y_3 \succ y_4 | x) \quad \Leftrightarrow \quad u(y_1, x) - u(y_2, x) \leq u(y_3, x) - u(y_4, x) \tag{35}$$

(and furthermore equality can only happen simultaneously on both sides). Consider the 2D plot giving $p(y \succ y' | x)$ as a function of $u(y, x) - u(y', x)$. (35) implies that it can be intrapolated by a strictly increasing function $F$. We also have $F(u(y, x) - u(y', x)) + F(u(y', x) - u(y, x)) = p(y \succ y' | x) + p(y' \succ y | x) \leq 1$ so $F(z) + F(-z) \leq 1$. This ends the proof of Theorem 4.2.

*Remark* B.  The crux of our proof consists in bypassing the use of the solvability axiom of the original proof (Krantz et al., 1989, Chapter 17, Theorem 2) – which is hard to motivate in the context of preference optimization – by creating a structure on top of alternatives / actions which can be (i) motivated in the context of preference optimization and (ii) compatible with the sampling process of preferences. We have resorted to simple – binary – lotteries, which have been motivated in a broad social choice context for decades (Machina, 1985). Other structures could be eligible. The major "modelling" price to pay is the definition of a local choice structure (LCS) that relies on properties that need to hold for all non-trivial $\alpha$s. This is a stark contrast with monotonicity, which just needs to hold for **one**. We could have alleviated considering all of $(0, 1)$ for the LCS by considering two open subsets of $[0, 1]$, one close to 0 and one close to 1. The formulation of a LCS we adopted is more restrictive but more readable, so we kept it.

## II.3. Proof of Theorem 4.3

The proof is obtained from the proof of two key results, the first of which is as follows (We remove the tilda notations to simplify readability since there is no misleading risk, e.g. with the loss definitions of step 1 in DPO or function $F$ in the KLST* structure).

**Theorem C.** *For any function $\phi$ satisfying (i) $\text{dom}\phi \supseteq [0,1]$, (ii) $\phi$ is strictly convex, the following loss $(\ell_0, \ell_1)$ is strictly proper*

$$\ell_1(p) \doteq -\phi(p) - (1-p)H(p) \quad , \quad \ell_0(p) \doteq -\phi(p) + pH(p), \tag{36}$$

*for any $H \in \partial\phi$. Furthermore if $|\phi(0)| \ll \infty, |\phi(1)| \ll \infty$, then we have the relationship $\ell_v(p) = D_{\phi,H}(v\|p) - \phi(v), \quad \forall v \in \{0,1\}$. Finally, if we add the constraint (iii) $\phi$ is symmetric around $x = 1/2$, then the loss as defined is symmetric.*

*Proof.* We want $-\phi(p) = p \cdot \ell_1(p) + (1-p) \cdot \ell_0(p)$ for some strictly proper loss $(\ell_0, \ell_1)$, that we assume wlog continuous on $(0,1)$ (because $\phi$ is necessarily continuous on $(0,1)$, being convex). We get the necessary relationship for $p < 1$:

$$\ell_0(p) = -\frac{\phi(p) + p\ell_1(p)}{1-p}. \tag{37}$$

Picking any $H \in \partial\phi$, the subdifferential of $L(p,q)$ at $q$, $\partial_q L(p,q)$, contains the following selections, using (37) to get rid of $\ell_0$:

$$\partial_q L(p,q) \ni pJ(q) - \frac{1-p}{(1-q)^2} \cdot ((H(q) + \ell(q) + qJ(q)) \cdot (1-q) + \phi(q) + q\ell(q)),$$

where $J$ is the derivative of $\ell_1$ where it is differentiable, and otherwise an element of the sub/super differential if it is not (a real in the interval defined by the left and right derivatives, guaranteed to exist because $\ell_1$ is continuous). The strict properness condition, that $\{0\} = \partial_q L(p,q)|_{p=q}$, imposes thus

$$pJ(p) - \frac{1-p}{(1-p)^2} \cdot ((H(p) + \ell_1(p) + pJ(p)) \cdot (1-p) + \phi(p) + p\ell_1(p)) = 0, \tag{38}$$

which simplifies in $\ell_1(p) = -\phi(p) - (1-p)H(p) = D_{\phi,H}(1\|p) - \phi(1)$ (if $\phi(1)$ is finite), as claimed. Using (37), we get

$$\ell_0(p) = -\frac{\phi(p) + p \cdot (-\phi(p) - (1-p)H(p))}{1-p}$$
$$= -\phi(p) + pH(p),$$

as claimed. The proof of Theorem C ends once we remind that a symmetric proper loss satisfies $\ell_0(p) = \ell_1(1-p)$ (Nock & Nielsen, 2008). $\square$

The proof of Theorem 4.3 relies on the following Lemma and second key result.

**Lemma D.** *For any function $\ell : [0,1] \to \overline{\mathbb{R}}$ strictly increasing, there exists $\ell_1 : [0,1] \to \overline{\mathbb{R}}$ such that $(\ell_0 \doteq \ell, \ell_1)$ is strictly proper. More specifically, we can pick, for any constants $K \in \mathbb{R}, a \in [0,1]$,*

$$(\ell_0, \ell_1)(p) = \left(\ell(p), K - \left(\int_a^p \frac{\ell(t)}{t^2}\,dt + \frac{1-p}{p} \cdot \ell(p)\right)\right)$$

*Proof.* Our proof is indeed constructive: we build $\ell_1$ via (36) by building

$$\ell \to \phi \to \ell_1 \tag{39}$$

We start by the left $\to$. Knowing $\ell, \phi$ in (36) is the solution of an ODE

$$y - py' = \ell$$

($\phi = -y$), from which we get the general solution ($K$ is a constant)

$$
\begin{aligned}
y &= K \exp\left(-\int \frac{1}{-t}\mathrm{d}t\right) + \exp\left(-\int \frac{1}{-t}\mathrm{d}t\right) \cdot \int \frac{\ell(t)}{-t} \cdot \exp\left(\int \frac{1}{-t}\mathrm{d}t\right) \mathrm{d}t \\
&= K \cdot p - p \cdot \int \frac{\ell(t)}{t^2}\mathrm{d}t.
\end{aligned}
$$

and we check that ($a \in [0, 1]$)

$$
\phi(p) = p \cdot \left(\int_a^p \frac{\ell(t)}{t^2}\mathrm{d}t - K\right)
$$

is indeed strictly convex [¶]. We then compute

$$
\begin{aligned}
H(p) &\doteq \phi'(p) \\
&= \int_a^p \frac{\ell(t)}{t^2}\mathrm{d}t + \frac{\ell(p)}{p} - K
\end{aligned}
$$

and, given that $\ell_0(p) \doteq \ell(p)$, we can complete (39) and get from (36)

$$
\begin{aligned}
\ell_1(p) &\doteq -\phi(p) - (1-p)H(p) \\
&= -p \cdot \left(\int_a^p \frac{\ell(t)}{t^2}\mathrm{d}t - K\right) - (1-p) \cdot \left(\int_a^p \frac{\ell(t)}{t^2}\mathrm{d}t + \frac{\ell(p)}{p} - K\right) \\
&= K - \left(\int_a^p \frac{\ell(t)}{t^2}\mathrm{d}t + \frac{1-p}{p} \cdot \ell(p)\right).
\end{aligned}
$$

This ends the proof of the Lemma. □

Remark that if $\ell$ is differentiable then $\ell_1$ is also differentiable and we get

$$
\ell_1'(p) = -\frac{1-p}{p} \cdot \ell'(p),
$$

which in addition to proving that $\ell_1$ is strictly decreasing also checks the properness condition when $(\ell_0 \doteq \ell, \ell_1)$ is differentiable (Reid & Williamson, 2010, Theorem 1). Now, $F : \mathbb{R} \to [0, 1]$ being strictly increasing, $F^{-1} : [0, 1] \to \mathbb{R}$ is also strictly increasing, so $\psi \circ F^{-1}$ is strictly increasing, and hence from Lemma D there exists a strictly proper loss $(\ell_0, \ell_1)$ such that $\ell_0 = \psi \circ F^{-1}$, or equivalently

$$
\psi = \ell_0 \circ F,
$$

as claimed. To prove the Theorem, we just have to remark that $\ell \doteq \psi \circ F^{-1}$ is a function $[0, 1] \to \mathbb{R}$ and strictly increasing, being the composition of two strictly increasing functions, which concludes the proof of Theorem 4.3 using Lemma D.

## II.4. Proof of Theorem 4.4

(We remove the tilda notations to simplify readability) Let us first show that

$$
\ell_0(p) - \ell_1(p) \in \partial\phi(p).
$$

This is equivalent to show that for any $q, p$,

$$
\phi(q) - \phi(p) - (q - p) \cdot (\ell_0(p) - \ell_1(p)) \geq 0.
$$

Using $\phi(v) \doteq -v\ell_1(v) - (1-v)\ell_0(v)$, this turns into the inequality

$$
-q\ell_1(q) - (1-q)\ell_0(q) + \underbrace{p\ell_1(p) + (1-p)\ell_0(p) - (q-p) \cdot (\ell_0(p) - \ell_1(p))}_{=q\ell_1(p)+(1-q)\ell_0(p)} \geq 0,
$$

---

[¶] If $\ell$ is differentiable, we get the simple expression $\phi''(p) = \ell'(p)/p > 0$ because $\ell$ is strictly increasing.

which becomes equivalently $q\ell_1(q) + (1-q)\ell_0(q) \leq q\ell_1(p) + (1-q)\ell_0(p)$, which is just the statement of properness for $\ell$ on $\Delta_2$ and thus shows $\ell_0(p) - \ell_1(p) \in \partial\phi(p)$. $\ell$ being strictly proper, $H(p) \doteq \ell_0(p) - \ell_1(p)$ is invertible so if we postulate $x = H(p)$, then necessarily

$$\ell_0(H^{-1}(x)) - \ell_1(H^{-1}(x)) \quad = \quad x.$$

Suppose $p \neq 0$, multiply both sides by $H^{-1}(x)$ and reorganize:

$$H^{-1}(x) \cdot \ell_0(H^{-1}(x)) \quad = \quad x \cdot H^{-1}(x) + H^{-1}(x) \cdot \ell_1(H^{-1}(x)).$$

Add $\ell_0(H^{-1}(x))$ to both sides and reorganize:

$$\begin{aligned}\ell_0(H^{-1}(x)) \quad &= \quad x \cdot H^{-1}(x) + H^{-1}(x) \cdot \ell_1(H^{-1}(x)) + (1 - H^{-1}(x)) \cdot \ell_0(H^{-1}(x)) \\ &= \quad x \cdot H^{-1}(x) - \phi(H^{-1}(x)) \end{aligned} \qquad (40)$$

(the last identity is the definition of $\phi$). Since $H \in \partial\phi(p)$, recovering $x = H(p)$ yields from the definition of the subdifferential that for any $t \in \text{dom}\phi$,

$$\phi(t) - \phi(p) - (t - p)H(p) \quad \geq \quad 0,$$

which reformulates as $t'x - \phi(t') \leq x \cdot H^{-1}(x) - \phi(H^{-1}(x))$ $(\forall t')$, implying $x \cdot H^{-1}(x) - \phi(H^{-1}(x)) = \phi^\star(x)$ for the right-hand side in (40), so we indeed get in the general case:

$$\ell_0 \circ H^{-1}(x) \quad = \quad \phi^\star(x).$$

Now, if the loss is symmetric, then it has $\ell_0(p) = \ell_1(1-p)$, inducing the symmetry $H(1-p) = -H(p)$, yielding

$$\ell_0(1 - H^{-1}(x)) \quad = \quad \ell_1 \circ H^{-1}(-x),$$

and so $\ell \circ H^{-1}(x) = \phi^\star(-x)$ (with the convention $\ell \doteq \ell_1$), as claimed.

## II.5. Proof of Theorem 5.1

With $\phi(\boldsymbol{p}) \doteq -\boldsymbol{p}^\top \boldsymbol{\ell}(\boldsymbol{p})$, the properness condition in $\boldsymbol{q}$ (4) is equivalent to having, $\forall \boldsymbol{p}, \boldsymbol{q} \in \Delta_n$,

$$\phi(\boldsymbol{p}) - \phi(\boldsymbol{q}) - (\boldsymbol{p} - \boldsymbol{q})^\top \cdot (-\boldsymbol{\ell}(\boldsymbol{q})) \quad \geq \quad 0. \qquad (41)$$

For any set $\{\boldsymbol{p}_1, \boldsymbol{p}_2, ..., \boldsymbol{p}_k\}$ $(k > 0)$ and $\boldsymbol{r} \in \Delta_k$, if we let $\boldsymbol{q} \doteq \mathbb{E}_{j \sim \boldsymbol{r}}[\boldsymbol{p}_j] \in \Delta_n$ and compute the expectation of the $k$ inequalities (41), we get

$$\mathbb{E}_{j \sim \boldsymbol{r}}[\phi(\boldsymbol{p}_j) - \phi(\boldsymbol{q}) - (\boldsymbol{p}_j - \boldsymbol{q})^\top \cdot (-\boldsymbol{\ell}(\boldsymbol{q}))] = \mathbb{E}_{j \sim \boldsymbol{r}}[\phi(\boldsymbol{p}_j)] - \phi(\mathbb{E}_{j \sim \boldsymbol{r}}[\boldsymbol{p}_j]) \quad \geq \quad 0,$$

so $\phi$ has to be convex on $\Delta_n$. The target $\boldsymbol{p}$ has three or more non-zero probabilities in indexes $\mathbb{I}$. In this case, if we restrict $\boldsymbol{q}$ to have non-zero coordinates in the same indexes (allowed by the properness assumption), then any two coordinates of $\boldsymbol{q}$ on these indexes are allowed to take values in a non-empty 2-dimensional open subset of a vector space; this, along with convexity and (41), impose the subdifferential property $-\boldsymbol{\ell} \in \partial\phi$ (without loss of generality, we suppose vectors restricted to indexes whose coordinates in $\boldsymbol{p}$ are non zero) which, since $\phi$ is separable and of the form $\phi(\boldsymbol{p}) = -\boldsymbol{p}^\top \boldsymbol{\ell}(\boldsymbol{p})$, provides at least 3 properties:

$$-\ell_i + K \quad \in \quad \partial - z\ell_i(z), \forall i \in \mathbb{I}, \qquad (42)$$

where $K$ is a constant. We note that for all these indexes

$$\partial - z\ell_i(z) \quad = \quad \{-\ell_i(z) - z \cdot h_i(z)\}_{h_i \in H_i}, \qquad (43)$$

where $H_i$ is the set of functions taking value $\ell_i'$ if it is differentiable or a real in the interval defined by the left and right derivatives if not. Hence (42) and (43) yield:

$$\exists K_1 \in \mathbb{R} : \forall i = 1, 2, ..., n, \exists h_i \in H : h_i(z) = -K_1/z.$$

$\ell_i$ being continuous (otherwise $\phi$ is not continuous, hence not convex), this implies $\ell_i(z) = -K_1 \log(z) + K_2$ $(\forall i \in \mathbb{I})$ with constraints $K_2 \in \mathbb{R}$ and $K_1 > 0$ (otherwise $\phi$ is not convex).

## II.6. Proof of Lemma 5.2

Showing the Lemma is a matter of writing

$$
\begin{aligned}
L(\boldsymbol{p}, \boldsymbol{q}) &= \sum_i p_i \ell_0(q_i) + p_i \cdot \sum_{j \neq i} \ell_1(1 - q_j) \\
&= \sum_i p_i \ell_0(q_i) + (1 - p_i)\ell_1(1 - q_i) \\
&\geq \sum_i p_i \ell_0(p_i) + (1 - p_i)\ell_1(1 - p_i) \\
&= L(\boldsymbol{p}, \boldsymbol{p}),
\end{aligned}
$$

where we have used $n$ times the properness of $(\ell_0, \ell_1)$ in the inequality (strict properness would involve a strict inequality). The last equality demonstrates the simplicity of the final loss in (22) ($L(\boldsymbol{p}, \boldsymbol{p}) = \sum_i p_i \ell_0(p_i) + (1 - p_i)\ell_1(1 - p_i)$), as exemplified by the binary entropy in ORPO (Hong et al., 2024).

## II.7. Proof of Lemma 5.3

Suppose $(\ell_0, \ell_1)$ is proper, i.e. $\forall p, q$ we have

$$
(1 - p)\ell_0(1 - p) + p\ell_1(p) \leq (1 - p)\ell_0(1 - q) + p\ell_1(q)
$$

Multiply both sides by $a > 0$, then add $c \cdot p$ and refactor, one gets:

$$
(1 - p)a \cdot \ell_0(1 - p) + p(a \cdot \ell_1(p) + c) \leq (1 - p)a \cdot \ell_0(1 - q) + p(a \cdot \ell_1(q) + c),
$$

i.e. $(\tilde{\ell}_0, \tilde{\ell}_1) \doteq (a\ell_0, a\ell_1 + c)$ is proper. The key quantities follow from standard convex optimization results, see e.g. Boyd & Vandenberghe (2004).

## II.8. Proof of Lemma 5.4

The solution to (24) follows from a standard property of Bregman divergences (see e.g. (Nock & Menon, 2020, Appendix)):

$$
\tilde{\pi}(y_i|x) = \phi'^{-1}\left(\frac{1}{n_i} \cdot \sum_k \phi'(\pi(y_{ik}|y_{i,<k}, x))\right). \tag{44}
$$

Remark the absence of Lagrange multipliers: such generalized means have the property that the solution $\tilde{\pi}(y_i|x)$ is between the min and max values of $\tilde{\pi}(y_{ik}|y_{i,<k}, x), j = 1, 2, ...$, so we are guaranteed that $\tilde{\pi}(y_i|x) \in [0, 1]$. The solution for the KL divergence is then immediate, the optimum being the geometric mean. For the Itakura-Saito (IS) divergence, we just have to rewrite the harmonic mean:

$$
\tilde{\pi}(y_i|x) = \frac{n_i}{\sum_k \frac{1}{\pi(y_{i,k}|y_{i,<k}, x)}} = \gamma_i \cdot \prod_k \pi(y_{i,k}|y_{i,<k}, x) = u(\pi(y_i|x)) \text{ for } u(z) \doteq z/\gamma_i,
$$

with $\gamma_i \doteq (1/n_i) \cdot \sum_l \prod_{k \neq l} \pi(y_{i,k}|y_{i,<k}, x) \ (\in [0, 1])$. It is hard to estimate the sum-product element but if we compute the $\alpha_i > 0$ such that

$$
\gamma_i = \exp(-\alpha_i n_i),
$$

then we get $u(z) = z \cdot \exp(\alpha_i n_i)$, as claimed. To finish up with the existence of $\alpha_i$, we show a precise interval

$$
\alpha_i \in [\underline{\alpha}_i, \overline{\alpha}_i]. \tag{45}
$$

To get interval (45) one just has to remark for example that

$$
\frac{1}{n_i} \cdot \sum_l \prod_{k \neq l} \pi(y_{i,k}|y_{i,<k}, x) \leq \frac{1}{n_i} \cdot n_i \cdot \max_k \pi(y_{i,k}|y_{i,<k}, x)^{n_i - 1},
$$

which after simplifying and taking logs leads to

$$\gamma_i \leq \exp\left(-\log\left(\frac{1}{\max_k \pi(y_{i,k}|y_{i,<k},x)}\right)\cdot(n_i-1)\right)$$
$$= \exp(-\overline{\alpha}_i \cdot n_i)$$

for

$$\overline{\alpha}_i \doteq \frac{n_i-1}{n_i}\cdot\log\left(\frac{1}{\max_k \pi(y_{i,k}|y_{i,<k},x)}\right).$$

Similarly, we get the expression for $\underline{\alpha}_i$:

$$\underline{\alpha}_i \doteq \frac{n_i-1}{n_i}\cdot\log\left(\frac{1}{\min_k \pi(y_{i,k}|y_{i,<k},x)}\right).$$

This ends the proof of Lemma 5.4.

*Remark* E. Note that $\underline{\alpha}_i, \overline{\alpha}_i$ depend on $n_i$, which may not be a desirable bias. To eliminate it, one just has to slightly correct the correction in (25) (main file) by the folllowing one:

$$\log\tilde{\pi}(y_i|x) = \log\pi(y_i|x) + \alpha_i(n_i-1).$$

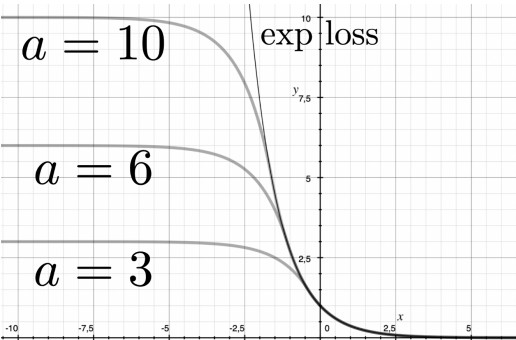

*Figure 4.* Plots of $\psi_a$ (46) for $a = 3, 6, 10$ (bottom-most to top-most thick curves) and the exponential loss (thin black curve).

## III. Supplementary material on toy experiment

From the standpoint of the choice of a loss function, two objectives eventually competing can be followed. The first is opt for Lipschitzness (e.g. logistic loss). In this case, one usually favors statistical consistency, the control of the Rademacher complexity, etc., and so focuses on generalization (Bartlett & Mendelson, 2002). Alternatively, one can opt for strong convexity (e.g. exponential loss over a right-bounded domain). In this case, one usually favors large curvature for fast/optimal convergence rate, and so rather focuses on training (Rakhlin et al., 2012). Our toy experiment, which is not meant in any way to be able to draw firm conclusions on any point discussed, consists in trying to balance both objectives via the tweaking of a loss which ends up being "far" from the DPO crowd because it is not convex anymore, yet fits under our normative umbrella.

Consider the exponential loss ($\psi(z) = \exp(-z)$). To balance the objectives above, we keep the exponential loss' strong convexity for margins ($z$ values) above a controllable threshold, and otherwise "bend it" further in the negatives to make it Lipschitz (while keeping its smoothness, convenient for optimization). Here is the resulting loss, non-convex, defined for $a \geq 2$:

$$\psi_a(z) \quad = \quad \begin{cases} a - \frac{a^2}{4} \cdot \exp(z) & \text{if } z < \log\left(\frac{2}{a}\right) \\ \exp(-z) & \text{otherwise} \end{cases} \tag{46}$$

Figure 4 plots three examples of $\psi_a$ for several values of $a$.

We use model GEMMA2_2B_IT, gradient clipping with global norm of 1.0. The maximum input length is set to 512. We use 1 H100 to run the experiment. Training dataset is Ultrafeedback-binarized. The model is trained for 200 steps with a batch size of 8, AdamW optimizer, peak learning rate of 1e-5, with a cosine lr scheduler and LR warm up for the first 10Eval config: model tested on Alpaca Eval v2. We use gemini-2.5-flash-lite as the rater.

