# OpenReview forum: "DPO Unchained: Your Training Algorithm is Secretly Disentangled in Human Choice Theory (and Its Loss' Convexity is Dispensable)"
_ICML.cc/2026/Conference — ICML 2026 spotlight_

### Official Review · Reviewer_T1bk · 2026-03-09

**Soundness:** 3
**Presentation:** 3
**Significance:** 3
**Originality:** 4
**Overall Recommendation:** 5
**Confidence:** 3

**Summary:**

This paper presents an ambitious theoretical generalization of DPO by expanding its normative foundation beyond the standard KL + sigmoid + logistic formulation. I appreciate the attempt to unify the RLHF objective, human choice modeling, and final loss under a broader proper-loss-based framework, which makes the paper intellectually interesting and potentially impactful. At the same time, I believe the paper would be further strengthened by clarifying its relationship to classical choice modeling foundations and by making the practical relevance of the generalized framework to existing DPO variants more explicit.

**Compliance With Llm Reviewing Policy:**

Affirmed.

**Final Justification:**

The rebuttal addressed my main concerns and reinforced my prior assessment.

**Key Questions For Authors:**

### Key Questions

1. **On abstention and the limitation of the BT model.**
   The paper emphasizes that the Bradley–Terry (BT) model cannot naturally handle *abstention*. However, in the broader choice modeling literature, there are already many models that incorporate a “leave” / “outside option” as an additional alternative with its own utility, and then model the final decision within the same utility-based framework. In this sense, it is not fully clear to me whether abstention is fundamentally beyond the BT/MNL family, or whether it can instead be absorbed into an augmented choice set. I would appreciate it if the authors could clarify more precisely what kind of abstention behavior is not captured by these existing utility-based formulations, and what is genuinely new in the proposed treatment.

2. **On the connection between the proposed generalization and Random Utility Theory (RUT).**
   My understanding is that BT, MNL, and more broadly logistic-style choice models are deeply connected to Random Utility Theory: the motivation is to make the probability that each option becomes the utility-maximizing alternative theoretically grounded and tractable, which leads to the introduction of Gumbel-type noise and eventually yields the familiar softmax / sigmoid structures. This is, in my view, a foundational design principle for choice models. The current generalization in the paper seems to abstract away from this perspective, and I think the paper would benefit greatly from explicitly clarifying the relationship between the proposed framework and the RUT view. In particular, is the proposed structure intended as a generalization of RUT-based choice models, an alternative normative foundation, or something orthogonal to them?

3. **On connecting the framework to common DPO variants.**
   I very much appreciate the authors’ effort to generalize DPO, and I agree with the broader message that the design space of DPO-style algorithms should extend well beyond the original formulation. To further strengthen the paper, it would be very helpful if the authors could analyze several representative DPO variants and explain how they fit into the proposed framework. Such discussion would make the theoretical contribution more concrete, and also help readers better understand the practical relevance of the framework for existing preference optimization methods.

**Limitations:**

Yes.

**Strengths And Weaknesses:**

### Strengths

- I appreciate the paper’s ambition in revisiting DPO from a more general normative perspective. The attempt to connect preference optimization with proper losses, generalized choice structures, and broader theoretical foundations is intellectually interesting and gives the paper a distinctive angle.

- The paper provides a fairly complete theoretical generalization of the original DPO pipeline, covering the RLHF objective, the human choice model, and the final loss. I find this decomposition clear and potentially valuable for understanding the broader design space of preference optimization methods.

- I also appreciate that the paper goes beyond a purely abstract formulation and tries to relate the theory back to practical algorithm design. The broader message — that DPO-style methods may admit a much richer design space than the standard KL + sigmoid + logistic recipe — is compelling and potentially useful for future work.

### Weaknesses

I refer the authors to the key questions part.

---

> ### Author Rebuttal · Authors · 2026-03-31
>
> (Note: bracketed expressions [reviewer_ID-letter] are used for easy cross reference between reviews)
>
> We thank the reviewer for assessing our work.
>
> As far as we know, our paper is the first attempt to capture in a normative framework combining both loss functions and human choice *the full extent of the DPO framework*, which consists in elevating the human choice model to the full correspondence with Savage’s properness. It is crucial to understand the potential extent of this normative “shield”, in particular given the applications on LLMs and their tremendous social impact (not to mention the potential price to pay to abandon this shield [9qcP-B]).
>
> # Key Questions
>
> > “[...] abstention and the limitation of BT [...]”
>
> **[T1bk-A]** In human decision theory or in applied econometrics, it is indeed standard practice to incorporate an outside option (see, for example [T’09] Chap. 2). In an economic choice context, modeling "no-choice" as additional option is justified by the fact that it yields an intrinsic/reservation utility. For instance, in a dynamic context (e.g. Search Theory), abstaining leaves open the opportunity to search for and select better options later on.
>
> However, the context in which we operate is different. Consider the set of responses generated by an LLM for a given prompt. This constitutes a highly heterogeneous choice space. Among these generated responses, certain pairs simply prove to be incomparable for a human evaluator.
>
> Thus, no-choice between two responses reflects a local incomparability rather than global rejection (i.e., if neither option reaches an absolute utility threshold).
>
> For this reason, we approach the choice set in a purely local manner (pairwise), which we then extend globally much like a graph: it contains chains of completely comparable options, but also pairs of incomparable options (resulting in abstention). The major advantage of this local architecture is analytical: rather than artificially complicating the general framework with a third global and intransitive utility which can be called `reservation utility’, it “absorbs” abstention behavior directly into the analytical properties of key functions involved, yielding new “freedom-of-design” choices (e.g. non-convexity), without requiring an additional no-choice option (Theorems 4.2, 4.3).
>
> > “[...] connection [...] Random Utility Theory [...]”
>
> **[T1bk-B]** We thank the reviewer for highlighting this fundamental connection to Random Utility Theory (RUT). To directly answer: our proposed structure is an alternative normative foundation that converges with standard RUT-based models under specific conditions, but ultimately acts as a generalization when the strict forced-choice assumption is relaxed.
> As the reviewer correctly points out, standard RUT relies on the assumption that the utility of an option is a random variable comprising a deterministic component and a stochastic error term ($U=V+\epsilon$) and, when these random variables are assumed to be independent and follow a specific distribution (e.g., Gumbel), the RUT framework naturally yields the familiar Bradley-Terry and MNL structures.
>
> Our framework reaches a compatible endpoint (the emergence of a strictly monotonic utility function) but it does so through a different theoretical path. Rather than postulating specific noise distributions *a priori*, we build upon the axiomatic foundations of stochastic choice [KLST’89]. Our Theorem 4.2 explicitly highlights this connection to the utility representation *without relying on explicit error terms*.
>
> A key advantage of this alternative axiomatic foundation is it achieves our objective *without* being constrained by the strict sum-to-one symmetry imposed by standard Gumbel-noise RUT models.
>
> We deliberately chose not to expand heavily on the standard RUT perspective. It can be the focus of a follow-up and our material is already quite dense for ML readers [dU11-E], yet we could use Appendix Section I to cover a bit of it.
>
> > “[...] connecting [...] DPO variants [...]”
>
> **[T1bk-C]** We agree with the reviewer’s suggestion. *However*, we faced a necessary compromise in using our paper’s space, between our key results (Sections 3 + 4) and consequences and directions (Sections 5 + 6). We managed to squeeze a small section (Section 5.5 + Table 1) related to the reviewer’s question, achieving a balance for which, given other reviewers comments [hKHC-C][9qcP-F], makes us think our paper landed in a relatively sweet spot regarding space optimization.
>
> Note: we have more material than Section 5.5, two more aggregation layers, including one more general encompassing e.g. [KLSL’25] that covers also [dU11-A]. We could push it to an Appendix.
>
> # References
>
> **[KLSL’25]** Kim et al.. “Spread preference annotation: Direct preference judgment for efficient LLM alignment”. ICLR’25.
>
> **[KLST’89]** Krantz et al., Foundations of Measurement Vol. 3, 1989
>
> **[T’09]** Train, Discrete Choice Methods with Simulation, 2009

---

> > ### Author Rebuttal · Reviewer_T1bk · 2026-04-01
> >
> > Thanks for the responses, I maintain my positive score.

---

> > > ### Author Response · Authors · 2026-04-03
> > >
> > > We would like to thank the reviewer for their response.

---

### Official Review · Reviewer_dU11 · 2026-03-09

**Soundness:** 3
**Presentation:** 2
**Significance:** 3
**Originality:** 3
**Overall Recommendation:** 4
**Confidence:** 2

**Summary:**

This paper generalizes the DPO framework by establishing an explicit connection with a normative model of human choice. The authors show that different choices of the two components in the algorithm, namely the reward function and the loss function, can correspond to different models of human decision making. Within this framework, the requirement of a convex loss function can be relaxed. In addition, the proposed normative perspective provides an explanation for several techniques used in existing DPO variants, such as margin-based formulations and length normalization.

**Compliance With Llm Reviewing Policy:**

Affirmed.

**Final Justification:**

The authors noted in their rebuttal that they will provide an explanation of the Machina lotteries in the Appendix and clarify how inconsistent or noisy responses impact the model. These proposed additions largely resolve my prior concerns and I have accordingly raised my score.

**Key Questions For Authors:**

1. If the zero abstention condition is imposed, could this introduce difficulties for preference learning when responses are inconsistent or contain noise? Some discussion of how the framework handles such situations would be helpful.
2. In Definition 3.2, could the authors clarify why $\textbf{ZA} \wedge \textbf{P} \Rightarrow \textbf{ZA}$ holds?
3. Using a nonconvex loss function may introduce optimization challenges, such as the presence of local minima. From the perspective of the proposed normative framework, what are the limitations of the standard DPO loss? In addition, what motivates the introduction of a nonconvex loss in this setting?
4. KTO (Ethayarajh et al., 2024) generalizes DPOusing prospect theory. Could the authors compare the proposed framework with KTO and discuss their similarities and differences?

**Limitations:**

Yes.

**Strengths And Weaknesses:**

Strengths:
1.  The paper shows that different choices of the two components in the algorithm, namely the reward function and the loss function, can correspond to different models of human decision making.
2. The proposed normative perspective provides an explanation for several techniques used in existing Direct Preference Optimization variants, such as margin-based formulations and length normalization, which is an interesting and novel perspective.
---
Weaknesses:
1. The normative framework proposed in this work is based on Machina lotteries, but this concept is not sufficiently explained in the paper. As a result, readers who are not familiar with this framework may find the presentation difficult to follow.
2. Definitions 3.2 and 3.3 are somewhat difficult to understand. Including illustrative figures or additional explanations in the main text may help improve clarity.

---

> ### Author Rebuttal · Authors · 2026-03-30
>
> (Note: bracketed expressions [reviewer_ID-letter] are used for easy cross reference between reviews)
>
> We thank the reviewer for assessing our work.
>
> # Weaknesses
>
> Explain Machina lotteries: we would be happy to provide in Appendix Section I, alongside [dU11-E][T1bk-B], simple pictures explaining those.
>
> # Key Questions
>
> > “[...] If the zero abstention [...], could this introduce difficulties for preference learning when responses are inconsistent or contain noise? [...] .”
>
> **[dU11-A]** Not tolerating abstention is *an issue with DPO’s Bradley-Terry-Luce (BTL) model*. Works have identified brittleness of DPO to flipped preferences [CKN24]. They change the logistic loss ($\psi$ (3) in our paper) to get more robustness. It turns out that *their new loss fits to our Theorem 4.3 if* the noisy reward difference is not too large compared to the noise level $\epsilon$ [dU11-H], which incidentally seems to hold in their IMDb experiments: the maximum noisy reward difference is 0.1 while for their chosen $\epsilon = 0.4$, we get $\log((1-\epsilon)/\epsilon) \approx 0.405 > 0.1$.
>
> In the context of our contribution, some lessons can be learned from this example:
>
> - **[dU11-B]** The paper [CKN24] is constrained by BTL’s straightjacket to improve PO with noise, but in fact it is not hard to show that *label smoothing techniques (an important class of techniques to mitigate noise) on proper losses could fit in our model* in the same way as [CKN24] does [dU11-H][LBMK20]. So our framework **could immediately safeguard** such techniques for a potential use in PO at low analytical cost [dU11-H];
>
> - **[dU11-C]** extracting valid new models of human choice to go beyond DPO, such as for better handling noise, could be tedious, in particular for the ML optimization/tuning loop (theory vs experiments) where the human choice model may "evolve"; our paper literally *gets rid of the need to extract the human choice model to carry out PO* as long as the ML part meets very weak assumptions (e.g. Theorem 4.3). In short, ML for PO can focus solely on… ML.
>
> - **[dU11-D]** Our paper considerably extends DPO’s BTL [9qcP-C, (i-b)] and would substantially alleviate constraints to connect human choice with ML-relevant models [PBR26];
>
> > In Definition 3.2, [...] why $ZA \wedge P \Rightarrow ZA$ holds?
>
> **[dU11-E]** We are not saying that $ZA \wedge P \Rightarrow ZA$ holds, but stating the definition of a LCS, so we assume the question is “why $ZA \wedge P \Rightarrow ZA$ **would** (realistically) hold?” (apologies otherwise). We provide an example in L155-L164 (col2) that we would be happy to present in simple graphical form in Appendix Section I. In short, *given our definition of preference* (7), if one **prefers** A against C and B against C *without abstention*, then there is knowledge (even implicit or unconscious) that breaks ties between A vs C and B vs C, from which it is reasonable to assume that this knowledge would “allow” to not abstain if/when presented with A vs B (note: no assumption on preference).
>
> > Using a nonconvex loss function [...] what motivates [...] a nonconvex loss in this setting?
>
> **[dU11-F]** Even when minimizing a convex loss *function* ($\psi$ (3) in our paper), the overall loss may not be convex in some model parameters, especially for deep architectures, so we do not “introduce” a complexity / challenge layer. However, Non-convexity brings an opportunity, exemplified in our *toy* experiment: the possibility to “flatten” strongly convex losses to control both curvature (influences convergence rate) and Lipshitzness (influences statistical consistency). Putting our *toy* experiment to full test would at least require a full paper.
>
> Finally, note that non-convexity is an additional **degree of freedom** authorized by our normative model.
>
> > KTO [...] similarities and differences?
>
> **[dU11-G]** A key difference is the type of data, where DPO-* approaches use “relative” (pairwise) preference data while KTO relies on “absolute” (pointwise) preferences, so models are different. Note also that (i) data is more often available as pairwise preferences and (ii) KTO suffers from “native” instability (both facts in the KTO paper).
>
> If the reviewer has any more questions, we would be happy to address them and thank them for reading our paper and rebuttals.
>
> # References
>
> [CKN24] Chowdhury et al., “Provably Robust DPO: Aligning Language Models with Noisy Feedback”, ICML’24
>
> [LBMK20] Lukasik et al., “Does Label Smoothing Mitigate Label Noise?”, ICML’20
>
> [PBR26] Pukdee et al., “What Does Preference Learning Recover from Pairwise Comparison Data ?”, ArXiv’26.
>
> # Appendix
>
> **[dU11-H]** the $\psi$ used in [CKN24] is $z \mapsto (1-\epsilon)\cdot \log(1+\exp(z)) + \epsilon\cdot \log(1+\exp(-z))$. For it to be strictly increasing, one needs $z \geq - \log((1-\epsilon)/\epsilon)$, which translates for the loss (which uses $\psi(-z)$) to the constraint $r(y,x) - r(y’,x) \leq \log((1-\epsilon)/\epsilon)$.

---

> > ### Author Rebuttal · Reviewer_dU11 · 2026-04-03
> >
> > Thanks for the detailed response. I increase the score to 4.

---

> > > ### Author Response · Authors · 2026-04-03
> > >
> > > We would like to thank the reviewer for their response and their decision to increase their score.

---

### Official Review · Reviewer_hKHC · 2026-03-13

**Soundness:** 4
**Presentation:** 3
**Significance:** 4
**Originality:** 4
**Overall Recommendation:** 5
**Confidence:** 2

**Summary:**

The present paper is a theoretical work about the foundations of Direct Preference Optimization (DPO) methods used in Reinforcement Learning from Human Feedback (RLHF). DPO is used in unsupervised large language models to better match human preferences by pushing models to produce responses towards what users want to see. The authors of the paper show DPO is disentangled in human choice and that some assumptions such as the necessity of convex losses are not essential. Instead of being part of an analytical framework, they reinterpret DPO features such as margins of length correction as elements of a normative one.

**Compliance With Llm Reviewing Policy:**

Affirmed.

**Key Questions For Authors:**

1.	Do the authors have any idea to provide concrete examples and quantitative metrics to evaluate the potential impact of this work? Is it possible to benchmark that work and apply it along datasets?
2.	Aside theory, what are the algorithmic and computational advantages of this work for concrete RLHF?
3.	The authors why losses should not necessarily be convex. Could they discuss about optimization stability and convergence problems potentially arising when training large models with such approaches?

**Limitations:**

The authors include an impact statement discussing potential implications of their theoretical framework, particularly regarding the role of normative assumptions in modeling human preferences.

**Strengths And Weaknesses:**

The paper provides a novel point of view on DPO using social choice theory allowing them to unify DPO features in the same theoretical framework. These ideas permit to connect different disciplines and could potentially lead to other connections. The point that convexity in losses are not necessities in preference optimization objectives is interesting. This conceptual work and the ideas presented are outstanding without any doubt. The main limitation of the paper (not the ideas) for me is that concrete examples are not provided, and no quantitative metrics are shown to evaluate the potential impact of this work.

---

> ### Author Rebuttal · Authors · 2026-03-30
>
> (Note: bracketed expressions [reviewer_ID-letter] are used for easy cross reference between reviews)
>
> We thank the reviewer for assessing our work.
>
> Since the Weakness part of the review is fully covered by key questions, we directly address them:
>
> # Key Questions
>
> > “[...] quantitative metrics to evaluate the potential impact of this work? [...] benchmark that work [...] ?”
>
> Fair point. Because our paper is the first paper so secure the normative part of preference optimization (PO) in its broadest sense (DPO did it in the narrower scope of the log/logistic loss for properness vs BTL for human choice), metrics to evaluate the impact of our work are both qualitative and quantitative. Our paper covers a subset of both:
>
> - **[hKHC-A]** how many existing approaches in “ordinary” PO setting (i.e. à-la-DPO) are automatically covered by our model (see also [T1bk-C]): Table 1 provides a list; rather a qualitative metric;
>
> - **[hKHC-B]** how many “variations” gravitating around “ordinary” PO could be derived using our model: this is Section 5.4 of our paper (see also [T1bk-C]); rather qualitative;
>
> - **[hKHC-C]** how many approaches to PO to **new settings** could be covered at minimal analytic cost, by *directly* using the corresponding techniques from supervised learning; consider the example of noise handling [dU11-A], still a poorly covered topic in PO. At almost zero analytical cost, existing approaches on label smoothing (label-flipping with fractional losses) could be covered at least for some noise regimes or maximal noise magnitude [dU11-B]. We insist on the minimal analytical cost [dU11-H]. Then a benchmark could be established, collecting all relevant label smoothing techniques (not just [CKN24]’s logistic loss) and comparing them. Obviously, this would require a paper of its own; the same could be said for outlier management, using off-the-shelf supervised techniques relying on a non-convex loss function [9qcP-C][CSWB06], since **crucially** non-convex loss functions are covered by our model. PO for Learning from Label Proportions could also be covered at minimal cost [PNCR14], with possible extension to privacy. Examples abound of ML settings where existing powerful supervised learning techniques could be extended to PO **at small analytical cost to show the normative protection of our human choice model**. Our *toy* example practices non-convexity for a simple problem, whose nice results could stem from several factors. Disentangling explanations would also require a paper of its own; includes both qualitative and quantitative metrics;
>
> - **[hKHC-D]** how many **new PO** approaches to “ordinary” PO setting (i.e. à-la-DPO) could be designed, using properties of our model that are absent from the DPO steam, such as non-convexity of the loss function, balances between strong convexity and Lipchitzness of the loss (learning rate vs consistency), or even using preference data where humans were allowed to abstain; includes both qualitative and quantitative metrics;
>
> - **[hKHC-E]** and last but not least, how many approaches would **avoid failing to be covered by our normative model thanks to our results** (in the course of their design). This one is not to be underestimated, given our Section 5.3 and Theorem 5.1. Because this would provide directions to *avoid* coverage failure before failure “happens”, this would rather be a “silent” (non measureable) metric. Or, interestingly, as we mention in conclusion, this would justify new avenues for human choice/loss functions [T1bk-B].
>
> > “[...] Aside theory, what are the algorithmic and computational advantages of this work for concrete RLHF? [...]”
>
> **[hKHC-F]** Fair question, and a simple answer: strong curvature (e.g. strong convexity) for the loss function favors fast convergence rates, for both GD and SGD. Many losses functions used in supervised learning to get fast rates would be immediately transferable and automatically get normative “protection” of our model of human choice (e.g. the exponential loss, that we use in our toy experiment).
>
> > “[...]The authors why losses should not necessarily be convex. [...] ?”
>
> Having a convex *loss function* does not prevent non-convexity in some model parameters (in particular for deep learning architectures), so issues of non-convexity due to the loss function do not dramatically change from those facing general learning with deep architectures. *However*, it is known that non-convexity of the *loss function* can be of help to handle data “issues” like outlier handling [9qcP-C][CSWB06]. Whether this can be extended to LLMs in the context of preference data remains to be proven (our toy experiment is in no way a proof, which would deserve a full paper), but it is definitely a very interesting problem given what is at stake with LLMs.
>
> # References
>
> [CSWB06] Collobert, Sinz, Weston, Bottou, “Trading convexity for scalability”, ICML’06
>
> [PNCR14] Patrini, Nock, Caetano, Rivera, “(Almost) No Label No Cry”, NeurIPS’14

---

> > ### Author Rebuttal · Reviewer_hKHC · 2026-04-02
> >
> > Thank you for the detailed rebuttal. The authors clarify several points and outline interesting directions for benchmarking and empirical evaluation. Follow-up question:
> >
> > - Do the authors modelised a minimal experimental setup to demonstrate one concrete advantage of their framework?

---

> > > ### Author Response · Authors · 2026-04-02
> > >
> > > We thank the reviewer for acknowledging our rebuttal and posting this question:
> > >
> > > The answer is yes, we have modelised a toy experiment. In the paper, this is Section 6. The section being small, we have provided additional details in Appendix, Section III.
> > >
> > > Before embarking as to why we chose this experiment vs another one, a word of caution: our paper developing from the ground up the normative model and its (many) properties and directions forward, we had a choice to make (space-wise) for presentation and it seems reasonable to think, from the reviewers’ reviews, that we achieved a balance between “more normative” (T1bk) and perhaps “more ML” (hKHC). Owing to the diversity of potential readers of our paper, we would love to keep this balance, but it means to keep space within bounds in the main file, even when we can of course expand in the Appendix.
> > >
> > > This being said, a bit more context into why we chose this experiment: it is well known that, from the standpoint of the choice of a loss function, two objectives eventually competing can be followed:
> > >
> > > - **(a)** opt for Lipschitzness (e.g. logistic loss). In this case, one usually favors statistical consistency, the control of the Rademacher complexity, etc., and so focuses on generalization [BM’02].
> > >
> > > - **(b)** opt for strong convexity (e.g. exponential loss over a right-bounded domain). In this case, one usually favors large curvature for fast/optimal convergence rate, and so focuses on training [RSS’12].
> > >
> > > From this point, our idea was simple: take a strongly convex loss, modify it so that we still have some good curvature (while losing on strong convexity), and gain a Lipschitz constant under control, while being smooth in all cases. A simple way to achieve this is via our simple clipping trick that breaks convexity but yields controllable curvature and Lipschitz constant. By venturing into non-convexity, we were also venturing in a region with still scarce approaches to preference optimization, but of course in the "safety basket" of our normative model.
> > >
> > > However, the choice of non-convexity may bring an additional advantage with potential benefits that would then need to be disentangled from (a) and (b):
> > >
> > > - **(c)** non-convexity can yield to better handling of outliers [hKHC-C][CSWB06]
> > >
> > > What we observe from our toy experiment: it does have potential, but to be fully covered, it would necessitate (at least) a full paper, with ideally not just experiments disentangling (a) vs (b) vs (c), but also theoretical consequences on the learning rates vs generalization vs outlier handling achieved. We are happy to leave this very interesting research question open until further work, and make those details more explicit in the Appendix, Section III.
> > >
> > > # References
> > >
> > > [BM’02] Bartlett, Mendelssohn, “Rademacher and Gaussian Complexities: Ricks bounds and Structural Results”, JMLR’02
> > >
> > > [RSS’12] Rakhlin, Shamir, Sridharan, “Making Gradient Descent Optimal for Strongly Convex Stochastic Optimization”, ICML’12

---

### Official Review · Reviewer_9qcP · 2026-03-14

**Soundness:** 3
**Presentation:** 3
**Significance:** 2
**Originality:** 2
**Overall Recommendation:** 4
**Confidence:** 2

**Summary:**

The paper shows that DPO is just one special case of a broader theoretical framework for preference optimization. The main result is that many different loss functions and choice models can be combined to produce valid preference-optimization algorithms, which means DPO occupies only a small part of a much larger design space. The paper also argues that preference optimization does not require convex losses or a specific human choice model.

**Compliance With Llm Reviewing Policy:**

Affirmed.

**Final Justification:**

My concerns are partially addressed by the authors' response, but I still have reservations regarding the necessity of the full choice-theoretic framework and the practical implications of the expanded design space. Overall, my assessment remains unchanged.

**Key Questions For Authors:**

To what extent is the full choice-theoretic framework necessary for deriving the main results, and could a similar conclusion be obtained purely from properties of proper losses?

**Limitations:**

yes

**Strengths And Weaknesses:**

**Strengths:**

The paper provides an interesting theoretical perspective on preference optimization and helps unify several DPO-style methods. By linking RLHF training with ideas from decision theory and proper loss functions, it clarifies the structure behind many algorithms and highlights a broader design space.

**Weaknesses:**

1. It is not clear how much of this theoretical generalization translates into practical improvements. While the framework shows that many losses and choice models are possible, the paper does not demonstrate that these alternatives lead to better training in real RLHF settings.

2. The framework also introduces quite a lot of machinery from choice theory, which may feel heavier than necessary given that the data in practice is just pairwise preferences. The final result suggests the specific choice model is not essential anyway, so the need for this detour is somewhat unclear.

3. While the paper expands the space of possible algorithms, there is little guidance on which choices might actually work well. The main result appears to be a reinterpretation of existing ideas about proper losses rather than a fundamentally new theoretical contribution.

---

> ### Author Rebuttal · Authors · 2026-03-30
>
> (Note: bracketed expressions [reviewer_ID-letter] are for easy cross reference between reviews)
>
> We thank the reviewer for assessing our work.
>
> # Weaknesses
>
> > “[...] how much [...] practical improvements. .[...]”
>
> Our paper is **the first normative approach on preference optimization (PO)**. We explain in four steps why this is crucial:
>
> **[9qcP-A] parallel with Savage’s seminal'71 paper**. Savage brought 2 major contributions for ML:
>
> - (i) **properness** guarantees the optimality of the data posterior = Bayes rule. So **we know what our model approximates** by optimizing a loss;
>
> - (ii) unveils the **analytical mechanism** behind properness (read: convexity/concavity/curvature of the loss function); it then becomes “easy” to design losses to address new problems and opportunities, safeguard existing approaches, new SOTA, etc. .
>
> **[9qcP-B] importance of our work in the context of (i)**. As messy as it is, reward modeling was framed in properness [CLBMLA97]. DPO’s brilliant, “Swiss watch” approach replaces this general normative guarantee by a specific one on human choice, Bradley-Terry-Luce (BTL). Most followers adopted one of two paths:
>
> - (i-a) stick to BTL, then becoming more of a design straightjacket (see e.g. [dU11-B]);
>
> - (i-b) **take the risk to abandon any link to human choice** (e.g. Khorasani et al.’ 25 in our refs). Because DPO’s normative guarantee on human choice theory replaces reward modelling’s guarantee from properness, abandoning it is a dangerous backward step for PO: it risks jeopardizing guarantees that training a LLM from human preference data can follow some model of human choice; *it is hard to exaggerate the potential dramatic consequences in the current context of LLMs, and it matters in the context of ML [SKD+’25]*;
>
> **[9qcP-C]** our contributions aligning with [9qcP-B]:
>
> - **In the context of (i-a)**, we free designs of PO from the need to comply with a specific model of human choice if the ML components meet weak constraints (Section 5.1). Read: no more straightjacket models of human choice and ML for PO can solely focus on… ML. [dU11-C]
>
> - **In the context of (i-b)**, our model of human choice is **much** broader than BTL and thereby can safeguard a larger variety of settings [dU11-A](Section 8).
>
> **[9qcP-D] importance of our work in the context of (ii)**. we unveil surprisingly lightweight conditions for ML compliance. In particular, relaxing the need for convexity of the loss function $\psi$ in (3) is important and counterintuitive in Savage’s context [9qcP-A].
>
> In this context, our contribution is threefold:
> - (1) “bring back in the safety basket” existing approaches with no known link to human choice [dU11-A],
> - (2) address new settings for PO for which supervised ML techniques abound, providing an easy “first benchmark” [dU11-B][hKHC-C],
> - (3) venture new ideas to improve SOTA, which is the reviewer’s remark. It should be clear at this point that it would have been out of the scope of our paper to make any form of extensive tests (a full paper would be necessary), the **toy** experiment we provide is a simple attempt at exploring non-convex losses, which can be useful (see e.g. [CSWB06][hKHC-C]).
>
> > “[...] lot of machinery [...] given that the data in practice is just pairwise preferences. [...]”
>
> **[9qcP-E]** misunderstanding: the data collected is simple *in description*, but the mechanisms involving human choice “behind the curtain” are tricky+ (7+ decades of history, see e.g. Econ Nobel Gérard Debreu'58 in our refs). Was it worth it to get to our level of details ? **Definitely yes**: reasons detailed in [9qcP-A][9qcP-B][9qcP-C][9qcP-D]. Complexity-wise, our machinery has been optimized [T1bk-A][T1bk-B].
>
> > “[...] choices might [...] work well. [...]”
>
> **[9qcP-F]** Our paper being the first "wide" normative PO approach, it is first focused on what **actually works** with respect to underlying normative assumptions [9qcP-A]. If “well” relates to applications, it would require more papers: our safety net for PO extends in ML to noise handling, outliers handling, learning from label proportions, among others [dU11-A][hKHC-C][T1bk-C].
>
> > “[...] reinterpretation of existing ideas about proper losses [...]”
>
> **[9qcP-G] factually incorrect**: properness is a *distributional/statistical property*. It tells nothing about mechanisms operating “behind the curtain” for (human) preferences (DPO’s BTL model, generalized to PO’s fullest extent in our context).
>
> # Key Questions
>
> > “[...] framework necessary [...] ?”
>
> See [9qcP-E][T1bk-A] as to why it is necessary.
>
> > “[...] similar conclusion [...] proper losses [...] ?”
>
> No, see [9qcP-G] for an explanation.
>
> If the reviewer has any more questions, we would be happy to address them.
>
> # References
>
> [CLBMLA97] Christiano et al.,“Deep Reinforcement Learning from Human Preferences”, NeuriPS’17
>
> [SKD+’25] Schaeffer et al.,“Position: Machine learning conferences should establish a ”refutations and critiques” track”. NeurIPS’25.

---

> > ### Author Rebuttal · Reviewer_9qcP · 2026-04-03
> >
> > Thank you for your response and clarifications. My concerns are partially addressed, but I still have reservations regarding the necessity of the full choice-theoretic framework and the practical implications of the expanded design space. That said, I appreciate the clarification of the intended normative scope of the work. Overall, my assessment remains unchanged and I do not have further questions at this stage.

---

### Decision · Program_Chairs · 2026-04-30

**Decision:**

Accept (spotlight)

**Comment:**

This paper introduces a comprehensive normative framework for DPO, demonstrating that preference optimization algorithms can operate successfully under a remarkably broad range of human choice models rather than being tightly restricted to the standard Bradley-Terry-Luce (BTL) model. By linking RLHF objectives with proper loss functions and the axiomatic foundations of stochastic choice theory, the authors prove that any compliant machine learning analytical choice can be embedded into the framework and that the convexity of the loss function is mathematically dispensable. This theoretical expansion unifies various existing DPO-style methods and opens up a significantly larger design space for developing future preference optimization algorithms. The reviewers find the results interesting with boarder impact. In the revised version, the authors should explicitly address the practical concerns raised by reviewers by providing clearer guidance or extended discussion on which non-convex losses and choice models might actually yield empirical improvements in real-world RLHF settings. Furthermore, the revised manuscript should ensure the heavy choice-theoretic machinery is well-justified within the maintext, clearly distinguishing why this extensive mathematical detour is necessary compared to simply relying on the properties of proper losses.